



# Supercooled liquid water cloud classification using lidar backscatter peak properties

Luke Whitehead[1], Adrian McDonald[1], and Adrien Guyot[2]

[1]School of Physical and Chemical Sciences, University of Canterbury, Christchurch, Aotearoa/New Zealand
[2]Australian Bureau of Meteorology, Melbourne, Victora, Australia

**Correspondence:** Adrian McDonald (adrian.mcdonald@canterbury.ac.nz)

**Abstract.** The use of depolarization lidar to measure atmospheric volume depolarization ratio (VDR) is a common technique to classify cloud phase (liquid or ice). Previous work using a machine learning framework, applied to peak properties derived from co-polarised attenuated backscatter data, has been demonstrated to effectively detect supercooled liquid water containing clouds (SLCC). However, the training data from Davis Station, Antarctica, includes no warm liquid water clouds (WLCC), potentially limiting the model's accuracy in regions where WLCC are present. In this work, we apply the Davis model to a 9-month Micro Pulse Lidar dataset collected in Christchurch, New Zealand, a location which includes WLCC. We then evaluate the results relative to a reference VDR cloud phase mask. We found that the Davis model performed relatively poorly at detecting SLCC with an accuracy of 0.62, often misclassifying WLCC as SLCC. We then trained a new model, using data from Christchurch, to perform SLCC detection on the same set of co-polarized attenuated backscatter peak properties. Our new model performed well, with accuracy scores as high as 0.89, highlighting the effectiveness of the machine learning technique when appropriate training data relevant to the location is used.

## 1 Introduction

Supercooled liquid water (SLW) droplets exist in clouds at temperatures below 0°C and above the homogeneous nucleation freezing temperature of around -40 °C (DeMott and Rogers, 1990). Heterogeneous nucleation of ice in clouds occurs when SLW droplets collide with ice nucleating particles (INPs) such as dust and other aerosols, or other ice particles (Khain and Pinsky, 2018). Regions in which INPs are scarce thus favour increased amounts of SLW cloud (Murray et al., 2012). In mixed-phase clouds between around -40 °C and 0 °C, SLW droplets and ice particles are both present. In this study, we refer to supercooled liquid water-containing clouds (SLCC) to describe clouds that could be mixed phase or SLW.

Previous work has shown that SLW and mixed-phase cloud are common over the Southern Hemisphere, particularly the Southern Ocean, using satellite (Hogan et al., 2004; Hu et al., 2010; Morrison et al., 2011; Huang et al., 2012) and in situ observations (Chubb et al., 2013), and that these clouds are under represented in climate models (Forbes and Ahlgrimm, 2014; Mason et al., 2015; Schuddeboom and McDonald, 2021). The important role of clouds in the Earth's energy balance means uncertainty in cloud occurrence and cloud phase has a large impact on models' radiation budgets (Bodas-Salcedo et al., 2016; Vergara-Temprado et al., 2018). When SLW clouds are under represented in models, too much sunlight warms the Southern



Ocean instead of being reflected from cloud tops back to space, causing an artificial heating. This has been shown to be the main contributor to sea surface temperature (SST) biases observed in many CMIP5 models (Hyder et al., 2018). Understanding the formation processes of supercooled liquid clouds is therefore an important topic of research to reduce biases in the radiation budget in global climate models over the Southern Hemisphere.

Moreover, recent work has shown that satellite-based measurements of low-altitude SLW cloud are prone to errors due to

attenuation from higher-altitude cloud (Blanchard et al., 2014; Protat et al., 2014; Liu et al., 2017; Alexander and Protat, 2018; McErlich et al., 2021). Therefore, while satellite-based measurements allow for global analysis of high-altitude SLW cloud occurrence, ground-based remote sensing observations are essential for accurate measurement of low-level clouds that are imperfectly measured from space.

Lidar is an active remote sensing technique that involves the emission of laser pulses into the atmosphere, and the mea-

surement of returned radiation backscatter from liquid drops, ice crystals, aerosols and other atmospheric constituents (Emeis, 2011). Automatic lidars and ceilometers (ALCs) are ground-based lidar instruments, as opposed to satellite-based instruments. Some ALCs, such as the MicroPulse Lidar used in this study, have depolarization capability that allows for the calculation of the linear volume depolarization ratio $\delta$, hereafter referred to as VDR, as calculated in Equation 1. This value includes contributions from both particle and molecular backscatter within a volume, and differs from the linear particle depolarization ratio

(Lewis et al., 2020). The utility of the VDR to determine cloud phase was first identified by Schotland et al. (1971) and has been used in numerous studies since to classify liquid water and ice-phase clouds (Sassen, 1991; Lewis et al., 2020; Ricaud et al., 2022). The physical principle behind this difference is that spherically symmetric liquid water droplets produce little to no depolarization, whereas backscatter from complex ice crystals tends to be depolarized and thus have higher VDR. Various studies have derived different thresholds to distinguish liquid- and ice-phase cloud, but most agree that $\delta < 0.1$ is characteristic

of liquid water clouds, with $\delta > 0.4$ for ice clouds and intermediate values representing mixed phase clouds (Sassen, 1991). It should be noted, however, that horizontally aligned ice crystals can produce specular reflections and decreased values of VDR, meaning such clouds can be falsely classified as liquid. This is usually mitigated by orienting the lidar off-zenith (Hogan and Illingworth, 2003). Furthermore, multiple scattering from multiple layers of liquid cloud can sometimes causes cross-polarized reflection and thus higher VDR, causing some liquid clouds to be falsely classified as ice.

Using only co-polarised backscatter, as measured by ceilometers, to detect SLCC would allow for SLCC occurrence to be analysed using widely-used existing ceilometer networks, negating the need for polarized lidar systems. Moreover, application to historical data sets would allow cloud phase retrieval to be extended to past records. Previous work by Hogan and Illingworth (1999) proposed a method of SLCC detection using ceilometers, and further studies developed new algorithms (Hogan et al., 2003; Hogan and O'Connor, 2004; Illingworth et al., 2007; Tuononen et al., 2019) for scientific and operational usage.

Illingworth et al. (2007) and Tuononen et al. (2019) used a SLCC classification scheme examining the attenuated backscatter coefficient profile and setting empirically-derived thresholds.

More recently, Guyot et al. (2022) implemented a data-driven classification model from lidar observations collected at Davis Station, Antarctica. A reference cloud phase mask was created from a merged depolarization lidar and W-band cloud radar product, and used to train an extreme gradient boosting (XGBoost) model (Chen and Guestrin, 2016) with single-polarization



ceilometer backscatter peak properties. The model was named G22-Davis to reflect that the training data was from Davis. Guyot et al. (2022) found that G22-Davis outperformed previous methods of SLCC detection, with accuracy scores as high as 0.91 compared to 0.84 for the application of the Tuononen et al. (2019) approach. A key consideration is that at Davis, virtually all liquid water will be in the supercooled state. No warm liquid water was detected over a year-long period based on ceilometer observations and the G22-Davis retrieval at Davis (Guyot et al., 2022). It is therefore important to determine whether the G22-Davis model can be applied to mid-latitude and lower latitude sites, where 'warm' liquid water clouds with temperatures greater than 0 °C exist. Furthermore, for the G22-Davis technique to be practical for wider use, it should be evaluated in a variety of conditions and regions. This provides the central motivation for this study.

The aims of this study are to: (i) evaluate the performance of G22-Davis for our dataset of MPL observations from Christchurch; (ii) using the same methodology as Guyot et al. (2022), develop a new model for SLCC classification trained using Christchurch MPL measurements; and (iii) apply the resulting cloud phase masks to produce a climatology of SLCC for Christchurch.

## 2 Data sets and Methodology

### 2.1 Data sets

#### 2.1.1 Christchurch MPL Observation Campaign

For the Christchurch field campaign, a Droplet Measurement Technologies Mini MicroPulse Lidar (MPL) was installed and operated from May 2021 to January 2022 on the roof of the Ernest Rutherford building on the University of Canterbury campus (43.5225°S, 172.5841°E) at an altitude of 45 m. The MPL is a compact dual-polarization elastic backscatter lidar that operates at a wavelength of 532 nm, and has a range of 30 km. For the Christchurch deployment, the vertical range resolution was set to 15 m, and the averaging time was set to 30 s. The scanning head of the MPL enclosure was set to a fixed vertical scanning mode with elevation angle 90°.

Post-processing of the raw MPL data is completed with version 1.2.1 of the Automatic Lidar and Ceilometer Framework (ALCF), a Linux software package detailed in Kuma et al. (2021b). While individual ceilometers typically implement post-processing in their firmware, ALCF allows a consistent noise-reduction and calibration method that can be applied to different ceilometer types, and has been applied in previous studies using ceilometer datasets in Antarctica (Guyot et al., 2022) and the Southern Ocean (Kremser et al., 2021; Pei et al., 2023). First, raw MPL data was converted with the *mpl2nc* tool (Kuma, 2020), which performs after-pulse, overlap and dead-time calibration and calculates cross- and co-polarized normalized relative backscatter (NRB) from cross- and co-polarized raw backscatter counts. ALCF performs noise reduction, absolute calibration and cloud detection from the *mpl2nc*-derived NRB, including resampling of the data to a common 5 minute temporal and 50 m vertical resolution. More details on ALCF methodologies are provided by Kuma et al. (2021b).



### 2.1.2 AMPS

The Antarctic Mesoscale Prediction System (AMPS) is a real-time limited area numerical weather prediction (NWP) model,
based on the Polar Weather Research and Forecasting (Polar WRF) model (Powers et al., 2012; Hines and Bromwich, 2008).
AMPS is used for scientific and logistical purposes in Antarctica, and extends to cover New Zealand because Christchurch is
a gateway city to Antarctica, providing access to Scott Base and McMurdo Station. The AMPS NZ grid has a 6 km spatial
resolution on 21 pressure levels, available in 3-hourly intervals initialised at 00:00 and 12:00 UTC. Real-time AMPS forecasts
are available online in GRIB1 format, and were set to automatically download during the study period and convert to NetCDF
(NC). Due to occasional download errors, AMPS data was not available for 29 days of the 9-month study period. Temperature
data was extracted for the nearest neighbour grid cell corresponding to the University of Canterbury site. The 2-dimensional
time × pressure level temperature field was cubic spline interpolated to a finer grid size and the hydrostatic balance equation,
assuming an isothermal atmosphere, was applied to resample to a 2-d time × altitude grid to match the resolution of the
ALCF-derived products. Isotherms at 0 °C and -40 °C were also determined from the AMPS output.

## 2.2 Cloud Phase Masks

ALCF performs cloud detection using an attenuated volume backscatter coefficient threshold algorithm (Kuma et al., 2021b).
The cloud mask was determined to be positive where the attenuated volume backscatter coefficient was greater than a tunable
threshold plus 5 standard deviations of noise. The default backscatter threshold for ALCF of $2 \times 10^{-6} \mathrm{m}^{-1} \mathrm{sr}^{-1}$ was used during
preliminary analysis of the Christchurch MPL dataset, but was found to be too low and resulted in a significant number of false
positive detections in the lowest 1 km likely due to the presence of boundary layer aerosols. Instead, a backscatter threshold of
$4 \times 10^{-6} \mathrm{m}^{-1} \mathrm{sr}^{-1}$ was chosen as a good compromise between boundary layer aerosol false positives and high altitude cirrus
false negatives based on visual inspection. The ALCF cloud mask product was used as the starting point of the depolarization
ratio reference mask.

### 2.2.1 MPL Depolarization Ratio Reference Mask

Several previous studies to determine cloud phase from polarized lidar backscatter retrievals use the linear volume depolarization ratio, VDR (Sassen, 1991). The principle is described in Sect. 1. VDR was calculated from *mpl2nc*-derived cross-polar
and co-polar normalised relative backscatter (NRB) using:

$$\delta(z) = \frac{P_\perp(z)}{P_\parallel(z)} \tag{1}$$

where $\delta(z)$ is the VDR at altitude $z$, and $P_\perp(z)$ and $P_\parallel(z)$ are the cross-polar and co-polar components of NRB respectively.

Since the *mpl2nc*-derived NRB products had not undergone re-sampling and noise reduction, VDR was sub-sampled to 50
m vertical and 5 minute temporal resolution to match the resolution of the ALCF products. In this way VDR, AMPS-derived
temperature and 2-dimensional ALCF products share a common time × altitude grid. VDR bins for which the corresponding





ALCF-derived cloud mask is 0 (i.e. no cloud) were set to not-a-number (NaN). To remove noise, values outside the range $\delta \in$
$(0,1)$ were also removed, and $\delta$ was averaged over consecutive cloud mask altitude bins to reduce speckle found in preliminary analysis. As per previous studies (Ricaud et al., 2022; Guyot et al., 2022), the threshold below which clouds were classified as liquid was set to $\delta_{\mathrm{LCC}} = 0.1$, where LCC refers to liquid water-containing cloud. Limitations to this simplification should be noted, especially associated with specular reflection from ice and multiple scattering leading to false classifications. The bin at altitude index $i$ and time step index $j$ with $\delta = \delta_{i,j}$ was then labelled LCC if $\delta_{i,j} \leq \delta_{\mathrm{LCC}}$ and ICC (ice-containing cloud)
if $\delta_{i,j} > \delta_{\mathrm{LCC}}$. If AMPS data was available, LCC-labelled bins were further classified as supercooled liquid water (SLCC) or warm liquid water containing clouds (WLCC) according to the AMPS-derived temperature at that bin, $T_{i,j}$. If $T_{i,j} \leq 0\ ^\circ\mathrm{C}$ that altitude bin was labelled SLCC and if $T_{i,j} > 0\ ^\circ\mathrm{C}$ the bin was labelled WLCC. We define SLCC as such because SLW and mixed-phased clouds cannot be distinguished using this method. However, WLCC must contain only liquid water, since ice cannot exist above $0\ ^\circ\mathrm{C}$. A profile at time step index $j$ was assigned a Boolean value $S_j$ for the presence of SLCC anywhere in
that profile. Likewise, $W_j$ and $I_j$ were assigned Boolean values for the presence of WLCC and ICC, respectively.

## 2.3    Development of a data-driven cloud phase mask

The method described here follows the process of Guyot et al. (2022) to develop a data-driven model for the classification of cloud as SLCC. The first step in the Guyot et al. (2022) methodology was to extract backscatter peak properties from a single-polarization ceilometer. In our case, ALCF-derived attenuated volume backscatter coefficient $\beta$ profiles from the MPL dataset
were used in place of the single-polarization ceilometer backscatter profiles. Since ALCF applies consistent calibration and resampling, the expectation is that the model developed here is applicable to any calibrated lidar data processed with ALCF, using the default 50 m vertical and 5 minute temporal resolution.

Each profile of attenuated volume backscatter coefficient $\beta$ was analysed using the signal processing tools of the Python library SciPy (Virtanen et al., 2020) to identify peaks (local maxima). Peaks exceeding a tunable minimum value of $\beta >$
$2 \times 10^{-5} \mathrm{m}^{-1}\mathrm{sr}^{-1}$ with minimum width of 50 m were found, and the following properties were recorded: the peak height $\beta\ (\mathrm{m}^{-1}\mathrm{sr}^{-1})$, the peak width $w$ (m), the peak width height $\beta_w\ (\mathrm{m}^{-1}\mathrm{sr}^{-1})$ defined as the height at which the peak width is evaluated, the peak prominence $\beta_{\mathrm{prom}}\ (\mathrm{m}^{-1}\mathrm{sr}^{-1})$ defined as the difference between the peak and the surrounding baseline, the peak altitude $z$ (m) above ground level, the total number of peaks for a given profile $n$ and the peak order within that total number in the range $(0, n)$. The peak width height was calculated as: $\beta_w = \beta - (0.5 \times \beta_{\mathrm{prom}})$. For a given profile with time step
index $j$, the corresponding AMPS-derived air temperature $T_{i,j}$ corresponding to the specific altitude bin $i$ of that peak, was also recorded. The 8-feature dataset of peak properties was then labelled with the reference mask classification of the altitude bin as 'SLCC', 'WLCC' or 'ICC', evaluated from the variables $S_{ij}$, $W_{ij}$ and $I_{ij}$.

Guyot et al. (2022) noted the importance of accounting for lidar extinction in multi-layer situations. Returned backscatter from higher altitude cloud where multiple layers are present will be weaker given the extinction of the lidar signal from
lower altitude cloud. They compared the value of peak backscatter for primary peaks (where peak order = 0) and secondary peaks (where peak order > 0), finding a clear separation between primary peak and secondary peak backscatter heights (Guyot et al., 2022, Figure 3). That work found peak height values of both primary and secondary peaks to have normal distributions





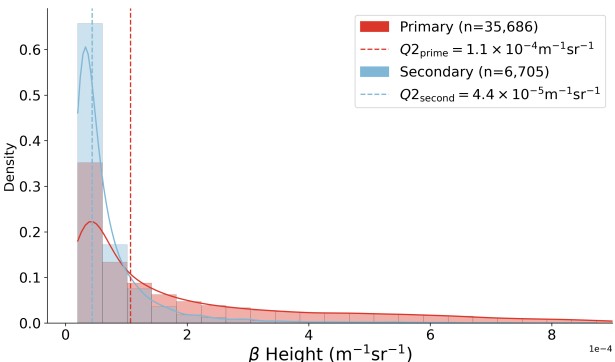

**Figure 1.** Histogram and kernel density estimation (KDE) plots of backscatter peak heights for primary (peak order = 0) and secondary (peak order > 0) peaks. Histograms are normalized such that the bar heights sum to 1. The median of each distribution, $Q2_{\text{prime}}$ and $Q2_{\text{second}}$ respectively, are also plotted.

and added an offset to the secondary peak heights, calculated as the absolute difference of the distributions' medians, for direct comparison. This offset of $4.2 \times 10^{-5} \text{m}^{-1}\text{sr}^{-1}$ was hypothesised to be the average extinction due to the presence of

the lower layer. We compared the distributions of primary and secondary peak properties for our dataset, which are shown in Fig. 1. Unlike Guyot et al. (2022, Figure 3), the peak height values are not normally distributed, and both skew right to greater values of attenuated backscatter peak height. That is, in our dataset there are more peaks with higher values of backscatter coefficient. This effect is more pronounced for primary peaks. This disparity from the distributions of Guyot et al. (2022, Figure 3) could be because our dataset is significantly larger (around 35,000 vs 3,700 primary peaks and 7,000 vs

570 secondary peaks) and therefore more varied; or because of an instrumental effect (e.g. due to the different wavelength of the Davis ceilometer, or a calibration difference); or due to a difference in environmental conditions (e.g. different aerosol concentrations causing more backscatter or more attenuation). The difference in the primary and secondary medians was calculated to be $6.6 \times 10^{-5} \text{m}^{-1}\text{sr}^{-1}$. As shown in Fig. 1, the backscatter height value corresponding to the maximum Kernel Density Estimate (KDE), i.e. the mode, is the same for primary and secondary peaks, and there is significant overlap between

the two distributions, unlike those in Guyot et al. (2022, Figure 3). Therefore, we chose not to scale the heights of secondary peaks by adding an offset.

    The next step in the Guyot et al. (2022) processing scheme was to train and test a data-driven model that could perform multi-class classification of each peak as SLCC, LCC or neither. As in Guyot et al. (2022), we also chose XGBoost due to its excellent performance in a wide range of applications (Chen and Guestrin, 2016), often outperforming other decision tree

or boosting model approaches. XGBoost is an optimised version of gradient tree boosting, an ensemble supervised learning algorithm. During training, XGBoost iteratively builds a series of 'weak' learners (regression trees) that are fitted to minimise the loss (i.e. error) of the resulting predictor, whilst also minimising complexity to avoid overfitting. XGBoost applies numerous performance optimisation strategies when building and combining the trees, reducing computational costs and allowing it to be scalable to large datasets. The XGBoost model developed by Guyot et al. (2022) performed binary classification on each



peak as SLCC or not. In this study, due to the presence of warm liquid water > 0 °C, we apply a multi-class classification of each peak as SLCC, WLCC or neither (implying the cloud phase is ice). The 8-feature dataset of peak properties was passed to the model for training along with the target label, which was the reference mask's classification of the peak as SLCC, WLCC or ICC. Model training and testing was performed on the entire peak dataset for which AMPS temperature data was available, excluding peaks from two case study days detailed later. The training dataset contained 36,767 peaks related to clouds from

223 days, of which 21% were labelled SLCC, 54% were labelled WLCC and 25% were labelled ICC by the VDR reference mask. Preliminary analysis on the peak dataset is presented in Section 3.1.

Data preparation and cross validation were implemented using the Python library scikit-learn (Pedregosa et al., 2011), and the XGB model was developed using the Python library XGBoost Chen and Guestrin (2016). To prevent overfitting we applied $k$-fold cross validation, in which the model is trained $k$ times on $k$ train/test folds. Test folds never share the same

data with other folds, allowing $k$ independent validation scores and preventing overfitting of the model to a specific training set. In the stratified $k$-fold cross validation used in this study, each training and testing fold contains approximately the same proportion of each target class as the complete set. Furthermore, group $k$-fold cross validation means grouped data is split into train/test folds such that the same group is not represented in both the training and test set. We used stratified group 3-fold cross validation. Groups were assigned by month, thus creating 9 groups with approximately equal size and class ratio. This

method meant neighbouring peaks were generally kept together, whilst it also preserved class ratios in the train/test sets. Due to the class imbalance in our peaks dataset (shown in Section 3.1) we used the balanced accuracy (described in more detail in Section 2.4) as the scoring method for cross validation and hyperparameter testing. Preliminary analysis to find optimal XGB hyperparameters was completed by applying an extended grid search, with 3-fold group stratified cross validation, over a range of maximum depth and learning rate ($\eta$) values. The depth of a regression tree is the number of splits (decisions) the tree

makes before reaching a prediction. Therefore the maximum depth hyperparameter controls how large a tree can grow, with larger values potentially improving predictive performance but increasing complexity. Learning rate is the 'shrinking' factor applied when trees are combined, and lower values reduce each tree's individual influence on the final prediction. Maximum depth values of [3, 6, 9] and learning rate values of [0.01, 0.02, 0.1, 0.3] were explored, and the optimal combination (max depth of 3 and learning rate of 0.1) was found to yield very little improvement in classification to any other hyperparameter

set: balanced accuracy scores increased by less than 0.01. Due to this insignificant improvement we continued using default XGBoost hyperparameters values (max depth of 6 and learning rate of 0.3).

The trained and tested XGBoost model was then applied to predict the classification for all peaks in the MPL dataset of attenuated backscatter. These model results were then used to create a cloud mask $Z_{ij}$ and $Z_j$ by processing each profile $j$ of attenuated backscatter sequentially: firstly, peak properties were calculated and recorded for the given profile. If no peaks

were detected, that profile was labelled $Z_j = 0$ or cloud free. The detected peaks' properties were passed to the trained model for classification as ICC, SLCC or WLCC. If the classification was SLCC or WLCC, the corresponding bin at peak altitude $Z_{ij}$ was labelled with that classification along with the surrounding bins, with the lower and upper bounds defined as twice the peak width value, as per Guyot et al. (2022). This created a cloud mask for detected layers of SLCC and WLCC. The ICC mask was determined from the cloud mask product from ALCF, with SLCC and WLCC layers subtracted. A profile $Z_j$ was





also labelled according to whether ICC, SLCC or WLCC were present anywhere in the profile. We hereafter refer to our trained
model as G22-Christchurch, to reflect that the model was trained on the Christchurch dataset. We applied the same method to
create the G22-Davis cloud mask evaluated in this study, for direct comparison with G22-Christchurch. Each peak was passed
to the G22-Davis model for classification as SLCC or ICC. If the classification was SLCC, the corresponding altitude bin was
labelled SLCC in the same process as described above. Again, the ICC mask was created by taking the ALCF cloud mask and
'subtracting' the SLCC layers.

## 2.4 Model performance metrics

Model testing and evaluation against the reference mask involves the comparison of two one-dimensional Boolean vectors
for each class (ICC, SLCC and WLCC). Here we provide basic definitions and describe metrics used for model evaluation in
this study, which are similar to those used by Guyot et al. (2022). A true positive (TP) is defined as a test result indicating
a correct prediction of a positive classification for a given class (e.g. presence of SLCC), and a true negative (TN) is defined
as a test result indicating a correct prediction of a negative classification (e.g. absence of SLCC). A false positive (FP) is a
test result indicating an incorrect prediction of a positive classification (e.g. wrongly predicting the presence of SLCC) and a
false negative (FN) is a test result indicating an incorrect prediction of a negative classification (e.g. incorrectly predicting the
absence of SLCC).

Recall, or true positive rate, is defined by Equation 2 and represents the proportion of all true samples that are correctly
classified as true, i.e. the ability of the classification to find all the positive samples:

$$\text{recall} = \frac{\text{TP}}{\text{TP} + \text{FN}} \tag{2}$$

Precision is defined by Equation 3 and represents the proportion of samples classified as true that are actually true, i.e. the
ability of the classification not to label as positive a negative sample:

$$\text{precision} = \frac{\text{TP}}{\text{TP} + \text{FP}} \tag{3}$$

The $f_1$ score is defined as the harmonic mean of the precision and recall defined in Equation 4:

$$f_1 = 2 \cdot \frac{\text{precision} \cdot \text{recall}}{\text{precision} + \text{recall}} \tag{4}$$

For the multi-class classification we apply in this study, precision, recall and $f_1$ scores are calculated for each class. Accuracy
is defined as the fraction of correct predictions out of the total number of samples. However, when classes are imbalanced (as
they are in this study), accuracy scores can be subject to inflated performance estimates. The balanced accuracy is defined as
the arithmetic mean of the recall obtained on each class, and is a more appropriate scoring method for imbalanced datasets
(Brodersen et al., 2010). In this study we primarily use balanced accuracy when describing the overall performance of the
classification, and $f_1$ scores when describing the performance for each class.





## 3 Results

### 3.1 Peak properties dataset

We first present analysis of the dataset of peak properties. The training dataset contained 36,767 peaks from 223 days, of which 21% (7,889 peaks) were labelled SLCC, 54% (19,822 peaks) were labelled WLCC and 25% (9,056 peaks) were labelled ICC by the VDR reference mask. In Fig. 2 we show kernel density estimation (KDE) plots showing the distribution of all the peak properties, separated by the reference mask's cloud phase classification as LCC (both SLCC and WLCC) or ICC. In Fig. 3 we show similar KDE plots of LCC peak properties, this time separated by the reference mask's classification of these LCC peaks as SLCC or WLCC.

Fig. 2a shows that values of backscatter peak height for LCC peaks are marginally higher than those for ICC peaks. The same is true for peak width height (Fig. 2c) and peak prominence (Fig. 2d), which also have units of $m^{-1}sr^{-1}$ and are strongly correlated with peak height, as we show later in Fig. 7. This supports our physical understanding that liquid water is associated with stronger backscatter returned signal, as found in previous studies Guyot et al. (2022). Though, clear overlap between these cases is observed in each set of distributions. It should also be noted that the difference in peak height between LCC and ICC peaks is small, and the distributions are poorly separated. Peak width, shown in Fig. 2b, also separates LCC and ICC peaks, with narrow peaks associated with LCC, and wider peaks associated with ICC. This is because liquid water attenuates the lidar signal more rapidly, and this attenuation is represented in the returned backscatter as a thin band. The number of peaks and the peak order, which are strongly correlated, show a slight separation between LCC and ICC peaks, where most LCC peaks are associated with a small number (1-2) of peaks in a single profile. This agrees with the findings of Guyot et al. (2022), who found that SLCC peaks tend to be associated with single-layer cloud most frequently. This is likely due to the attenuation effects of LCC layers that obscure the view of higher altitude cloud in multi-layer situations.

Fig. 2e shows the altitude distribution of LCC and ICC peaks. It shows that LCC peaks are most strongly associated with lower altitudes, with the frequency decreasing as altitude increases. The temperature distribution in Fig. 2f shows the inverse is true of temperature: that LCC peaks are most strongly associated with warmer temperatures as we expect, and the frequency of LCC peaks decreases as temperature decreases until the lower temperature limit of around -40 °C is reached, below which homogeneous freezing occurs (DeMott and Rogers, 1990). Fig. 2f shows that ice peaks are present at temperatures as low as -70 °C, and also above 0 °C. However, it is likely that this 'warm' ICC is actually LCC that was misclassified by the reference mask as ICC due to multiple scattering. It should be reiterated that the AMPS temperature information was only used in the VDR reference mask to distinguish SLCC and WLCC. Temperature was not used to discriminate LCC and ICC, and only the volume depolarization ratio $\delta$ was used for that separation. Overall, the temperature distribution in Fig. 2f is representative of SLCC temperature distributions in the heterogeneous freezing temperature regime as identified in previous work (Murray et al., 2012; McErlich et al., 2021). This gives us confidence that our VDR reference mask is accurately distinguishing SLCC and ICC in the heterogeneous temperature range.

Fig. 3 shows the difference in average peak properties between peaks classified as SLCC and WLCC by the reference mask. Fig. 3a shows that WLCC peaks generally have higher values of backscatter height than SLCC peaks, and the same is true



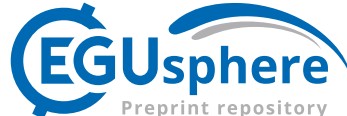

**Figure 2.** Kernel density estimation (KDE) plots of peak property distributions for the full dataset, with the distribution's median also plotted. Peaks are separated by the reference mask's cloud phase classification as LCC or ICC. For n Peaks and Peak Order (g, h), histograms and median lines are omitted for clarity. The unit for peak width is a number of range-gates, which can be converted to distance by multiplying by 50 m.



**Figure 3.** Kernel density estimation (KDE) plots of peak property distributions for the full dataset, with the distribution's median also plotted. Peaks are separated by the reference mask's cloud phase classification as SLCC or WLCC. For n Peaks and Peak Order (g, h), histograms and median lines are omitted for clarity.



for peak width height and prominence. However, this result was found when looking at peaks from all altitudes. Given that WLCC peaks are nearly always found at altitudes below 2 km, and SLCC peaks are always found above 0.5 km (as shown in

Fig. 3e), any altitude-dependent bias in attenuated volume backscatter would carry over to the SLCC and WLCC backscatter height distribution. A fair comparison would require all peaks to be in the same altitude range. By comparing the average peak properties over smaller altitude ranges (below 0.5 km, 0.5-1 km, 1-1.5 km, 1.5-2 km and above 2 km), we found that the distributions of peak height were roughly equal and similarly skewed between SLCC and WLCC peaks. It is possible, then, that an altitude-dependent bias exists in the attenuated backscatter profile. This could be caused by an imperfect overlap or

range correction in the lidar processing. However, our analysis found that this bias made little difference to the performance of the G22-Christchurch model and cloud mask.

As expected, the temperature distribution shown in Fig. 3f shows that SLCC peaks have temperatures between around -40 °C and 0 °C, and WLCC peaks have temperatures above 0 °C. The altitude distribution shown in Fig. 3e shows that WLCC peaks are most frequently found in altitudes below 2 km, and that SLCC peaks are found between around 0.5 km and 8 km.

The overlap in the distributions of SLCC and WLCC peak altitude between 0.5-2 km appears to be due to the seasonal and daily variation in the altitude of the 0 °C isotherm.

## 3.2 Model performance evaluation

The XGBoost model was trained and tested with 3-fold stratified group cross-validation, as discussed previously. Accuracy scores, as described in Section 2.4, are presented here as the mean of the 3-fold training and testing scores, with the uncertainties

derived from the standard deviation. The balanced accuracy score was $0.872 \pm 0.018$, and the f1 scores for each class were: $0.788 \pm 0.035$ for ICC, $0.847 \pm 0.022$ for SLCC and $0.980 \pm 0.003$ for WLCC for the G22-Christchurch analysis. We also applied G22-Davis to our dataset of peak properties, and compared that model's binary prediction to our reference mask's classification of each peak as either SLCC or LCC. Compared to the reference mask's SLCC classification, the G22-Davis balanced accuracy score was only $0.41$ and the f1 score for the SLCC class (positive binary classification) was $0.13$. Compared to the reference

mask's LCC classification, the G22-Davis balanced accuracy score was $0.66$ with f1 score $0.54$ for LCC (i.e. both SLCC and WLCC). These results demonstrate that the G22-Davis model performed poorly at SLCC and LCC classification in this environment relative to the excellent performance at Davis, where the model achieved accuracy scores as high as $0.91$ (Guyot et al., 2022).

The performance of the SLCC mask was then analysed by comparing the reference mask's SLCC label $S_j$ to the model's

SLCC mask label $Z_j$. Balanced accuracy scores were again derived, given the class imbalance. We first evaluate the performance of the model mask by comparing all profiles (time steps), and then compare only time steps where peaks meeting the minimum backscatter and width thresholds were detected. For the full dataset with 66,240 time steps, there were 31,708 time steps (47.9%) where peaks were detected. This is lower than the total number of peaks (36,767) since some time steps contained multiple peaks corresponding to multi-layer cloud. According to the reference mask, SLCC was present in 14.2% of all

profiles, and 25.4% of profiles with peaks. The balanced accuracy of G22-Christchurch SLCC mask was $0.85$ for all profiles, and $0.89$ for profiles with peaks. The balanced accuracy for the G22-Davis SLCC mask, compared to the reference mask, was





only 0.62 for all profiles, and 0.55 for profiles with peaks. We repeated this analysis for WLCC detection by comparing the reference mask's WLCC label $W_j$ to our model's WLCC mask. According to the reference mask, WLCC was present in 44.6% of all profiles, and 66.1% of profiles where peaks were detected. The balanced accuracy of the G22-Christchurch WLCC mask was 0.73 for all profiles, and 0.81 for profiles with peaks.

## 3.3 Case studies

In this section we present results showing the application of both our mask and the G22-Davis mask to two case studies, in order to interpret their performance. Backscatter peaks from these days were excluded from the training dataset, so these days represent unseen data.

### 3.3.1 2021-05-18 Case study

Observations from 18 May 2021 are shown in Fig. 4 as an example of a day with a distinct LCC layer. Fig. 4 shows the ALCF-calibrated attenuated volume backscatter (Fig. 4a), volume depolarization ratio VDR (Fig. 4b) and VDR reference cloud mask (Fig. 4c-d), along with the G22-Davis cloud mask (Fig. 4e-f) and G22-Christchurch cloud mask (Fig. 4g-h) applied to the attenuated backscatter profiles. The cloud masks are presented as time × altitude grids $Z_{ij}$ (Fig. 4c, e, g) and time-step Booleans $Z_j$ (Fig. 4d, f, h) according to whether SLCC is present anywhere in that profile. This particular day shows a clear band of SLCC between 06:00 and 12:00 UTC at 4km altitude, and again between 17:00 18/05 and 00:00 19/05 UTC between 3-6km. Other cloud (identified as ice by the VDR reference mask) is present from 02:30 to 06:00 UTC between 2-8km altitude, and again from 14:30 to 16:30 UTC. The first SLCC band is a multi-layered cloud region between 06:00 and 08:00 UTC. Clouds or precipitation can be observed below the second SLCC band at around 20:00 and 22:00 UTC. A thin layer of WLCC below 1 km is also present from 19:00 UTC 18/05 to 00:00 UTC 19/05.

Both G22-Davis and our model's cloud masks correctly identify the second SLCC band. The first SLCC band, which appears slightly thicker, is poorly represented in the G22-Davis cloud mask, but correctly identified using our model. Comparing profile-to-profile SLCC classification (i.e. comparing Figs. 4d, f, h), G22-Davis had a balanced accuracy score of 0.76 for all profiles, and 0.68 for profiles containing peaks. Our model performed better with balanced accuracy scores of 0.93 for all profiles and 0.84 for profiles containing peaks, for this case study.

### 3.3.2 2021-06-05 Case study

MPL observations from 5 June 2021 are shown in Fig. 5. On this day, a thick band of ice cloud is present at altitudes ranging from 3-10 km between 00:00 and 18:00 UTC, followed by a band of low-altitude (< 1 km) liquid water cloud between 18:00 UTC 05/06 and 00:00 UTC 06/06, and another layer of liquid water cloud at around 2-3 km altitude for a short period after 22:00 UTC. Layers of SLCC are present for short periods throughout the day (06:00, 10:00, 12:00 and 14:00 UTC) interspersed with ice cloud. Both G22-Davis and our model identify some of the short periods of SLCC between 06:00 and 15:00 UTC. Our model also correctly distinguishes most of the SLCC, but also overestimates SLCC occurrence, making false positive



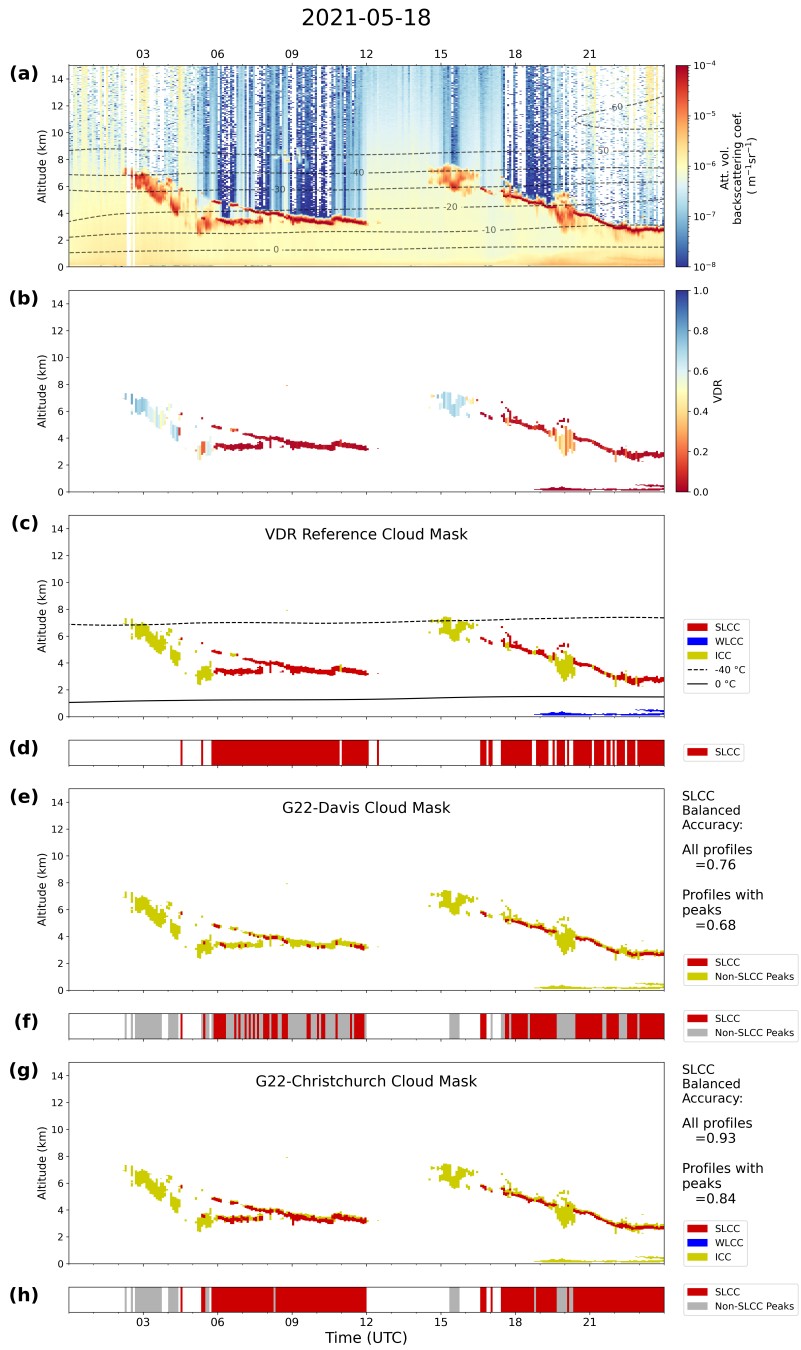

**Figure 4.** MPL profile for 2021-05-18 over Christchurch showing attenuated volume backscatter coefficient **(a)**, volume depolarization ratio VDR **(b)**, the VDR reference cloud mask **(c)**, the G22-Davis cloud mask **(e)** and the G22-Christchurch cloud mask **(g)**. Time-step classifications of SLCC are shown in **(d, f, h)** for the reference mask, G22-Davis and G22-Christchurch respectively, along with time steps for which peaks were identified. Also shown are balanced accuracy scores for G22-Davis and G22-Christchurch. AMPS air temperature contours are overlaid in **(a)** and **(c)**.

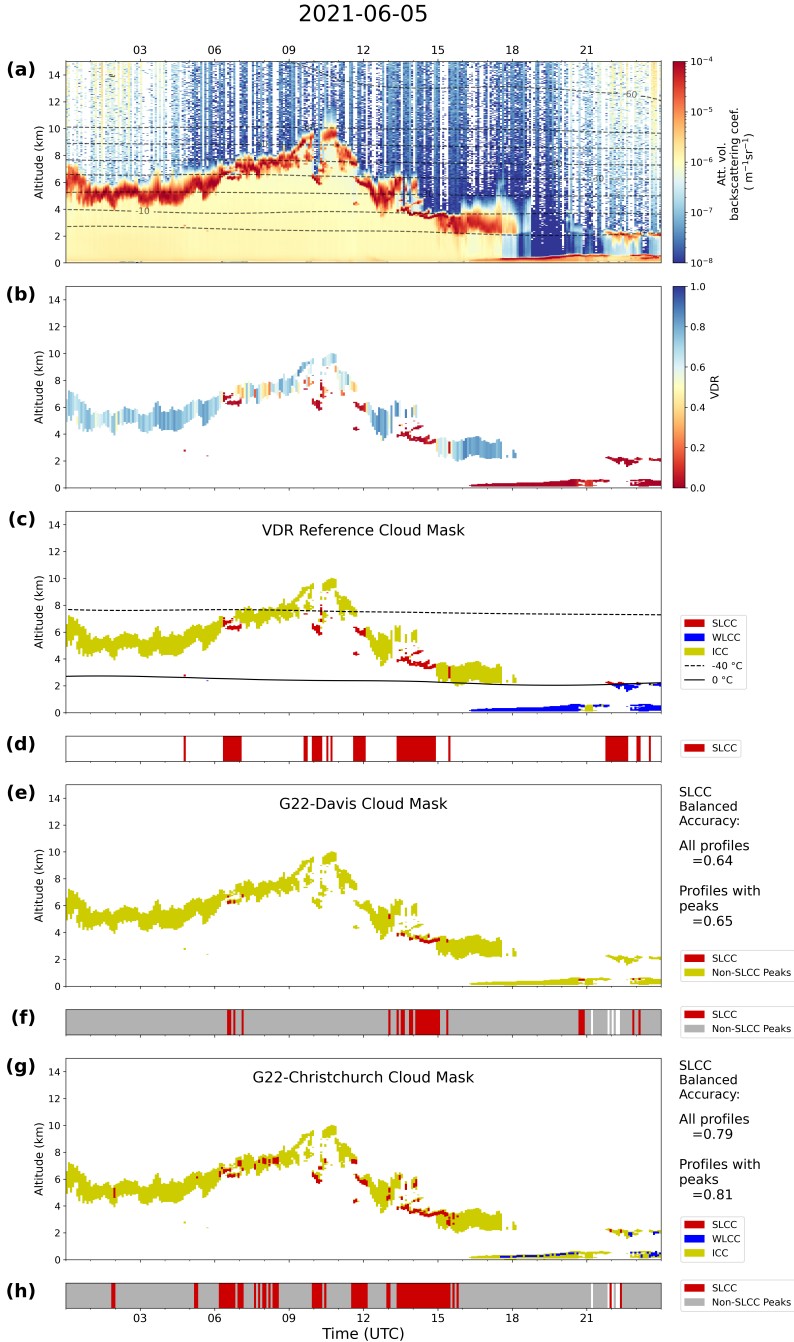

**Figure 5.** MPL profile for 2021-06-05 over Christchurch showing attenuated volume backscatter coefficient **(a)**, volume depolarization ratio VDR **(b)**, the VDR reference cloud mask **(c)**, the G22-Davis cloud mask **(e)** and the G22-Christchurch cloud mask **(g)**. Time-step classifications of SLCC are shown in **(d, f, h)** for the reference mask, G22-Davis and G22-Christchurch respectively, along with time steps for which peaks were identified. Also shown are balanced accuracy scores for G22-Davis and G22-Christchurch. AMPS air temperature contours are overlaid in **(a)** and **(c)**.





classifications at around 07:00, 09:00 and 14:00 UTC. G22-Davis does not identify the low altitude WLCC between 18:00
UTC 05/06 and 00:00 UTC 06/06 as SLCC. Our model correctly identifies some of this layer as WLCC, although much of
the layer is missed. For this case study, G22-Davis had a balanced accuracy score of 0.64 when comparing all profiles, and
0.65 when comparing profiles containing peaks. Again, our model performed better, with a balanced accuracy of 0.79 for all
profiles and 0.81 for profiles containing peaks. Peaks were detected in 98% of the profiles for this day, hence the similarities
between scores for all profiles and profiles with peaks. These results show that our model performed very well, and indicates
that the technique of Guyot et al. (2022) can be applied successfully to a new location if appropriate site-specific training data
is available.

### 3.4 Cloud occurrence for the full observation period

The full dataset was analysed and cloud occurrence statistics computed to compare the accuracy of the G22-Davis and G22-
Christchurch cloud masks to our VDR reference mask. In this section, we present cloud fraction statistics and cloud phase
distribution as a function of altitude for each mask, and evaluate the performance of the G22-Christchurch cloud mask.

Cloud fraction was calculated by finding the proportion of profiles in which SLCC or WLCC was detected, across all time
steps of the dataset. The cloud fraction from all types (calculated from the ALCF cloud detection mask) was 70% for the full
dataset of 257 equivalent days of MPL profiles. SLCC was detected in 14.2% of the reference mask profiles, 12.2% of the
G22-Christchurch mask profiles and 17.3% of the G22-Davis mask profiles. Meanwhile, WLCC frequency was 44.6% for the
reference mask and 20.5% for G22-Christchurch. As we discuss later in this section, G22-Davis overestimated the frequency of
SLCC occurrence because it often misclassified WLCC layers as SLCC at this mid-latitude site. G22-Christchurch, however,
tends to underestimate SLCC and WLCC occurrence, although the WLCC underestimation is more significant. From visual
analysis of WLCC layers, such as the low altitude layer between 18:00 and 21:00 UTC in Fig. 5, we found that peaks were
occasionally misclassified by G22-Christchurch as non-WLCC, leading to a 'speckle' pattern and over-representation of ICC
in these layers. Because these layers have temperatures greater than 0 °C (shown by the relative position of the 0 °C isotherm),
we would naturally expect them to be warm liquid water, by definition. Therefore, it is a shortcoming of the G22-Christchurch
model that ICC is detected in this temperature region. However, sometimes the VDR reference mask also detected ICC in this
temperature region. This might be attributed to multiple scattering in WLCC layers having the effect of increasing VDR and
thus causing classification as ICC. In the example in Fig. 5c, a short period of 'ICC' is present at 21:00 UTC. From Fig. 5b,
however, this cloud has a VDR value of around $\delta = 0.15$, only slightly higher than the $\delta_{\mathrm{LCC}} = 0.1$ threshold. This demonstrates
the sensitivity of the reference mask, and thus the G22-Christchurch model, to the VDR threshold $\delta_{\mathrm{LCC}}$, and suggests that $\delta_{\mathrm{LCC}}$
should be increased. It is therefore possible that the G22-Christchurch model erroneously learnt, from the reference mask, that
ICC could be present at warm temperatures and low altitudes.

Another potential reason for the underestimation of WLCC and overestimation of ICC in low-level cloud by the G22-
Christchurch cloud mask is that the method for creating the cloud mask from the classified peaks is potentially sub-optimal.
As described in Section 2.3, we defined ICC based on the ALCF-derived cloud mask with SLCC and WLCC layers subtracted.
This is similar to the method used by Guyot et al. (2022), in which the ice cloud mask is derived using the method detailed in



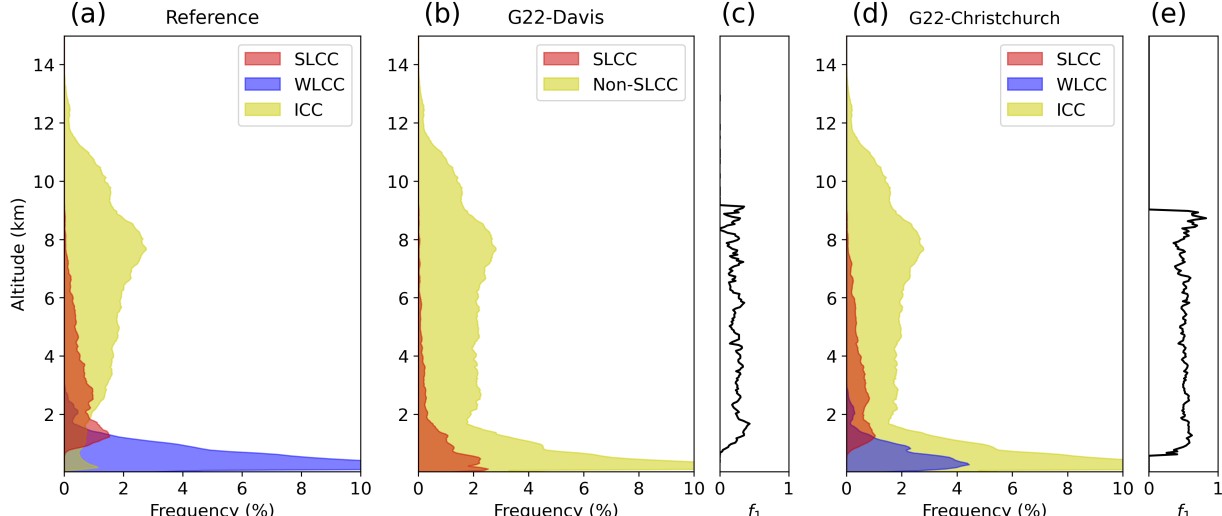

**Figure 6.** Cloud altitude distributions separated by cloud phase, according to the VDR reference mask **(a)**, G22-Davis mask **(b)** and G22-Christchurch mask **(d)**. $f_1$ accuracy scores for SLCC detection, as a function of altitude, are also shown for G22-Davis **(c)** and G22-Christchurch **(e)**.

Tuononen et al. (2019). For the G22-Christchurch mask, in the SLCC temperature region, non-SLCC layers are unambiguously labelled ICC. However, in the WLCC temperature region, non-WLCC layers are labelled ICC, even though ice cannot exist in temperatures greater than 0 °C. This may thus lead to mislabelling of precipitation and fog. This is demonstrated once again
in Fig. 5g in the low altitude WLCC layer between 18:00 and 21:00 UTC. The G22-Christchurch mask erroneously identifies a large amount of this cloud as ICC. This is due, in part, to the model incorrectly classifying some peaks as ICC, as discussed previously. However, other parts of the cloud are identified as ICC by the mask because there are either no peaks detected (so the cloudy bins in that profile are conservatively labelled ICC by default) or because the cloudy bins above or below a WLCC peak are labelled ICC.

Fig. 6 compares the cloud phase distributions as a function of altitude for G22-Davis and our model against the reference mask. According to the reference mask Fig. 6a, LCC is common at low altitudes, and decreases in frequency with altitude to a maximum altitude of 8 km. Below 1 km, this LCC is entirely WLCC, and above 3 km it is entirely SLCC, with the overlap between 1-3 km likely corresponding to the variation of the 0 °C isotherm through the year. This pattern is consistent with the peak altitude distributions shown in Figs. 2e and 3e. Ice cloud frequency increases with altitude to a maximum at 8 km, before
decreasing to very low occurrences at 13 km. The reference mask also shows a low frequency of ice cloud occurrence at low altitudes, which is also present in the peak altitude distribution in Fig. 2e. However, it seems likely that this is LCC that has been misclassified as ICC due to the effects of multiple scattering, as discussed earlier in this section.





Fig. 6b shows that G22-Davis identifies SLCC at lower altitudes (0-4 km) than the reference mask (1-8 km). The $f_1$ score as a function of altitude, in Fig 6c, is relatively low and reduces to zero in the lowest 2 km for this reason. This error is likely

because the model learnt from the Davis training dataset that SLCC could be present at low altitudes, despite the fact that temperatures for which SLCC was detected were much lower. This therefore provides evidence of the limitation of applying the G22-Davis method, with training data from Davis, Antarctica, to mid-latitude sites with warmer air temperatures. This also shows the relative importance of peak altitude in the SLCC classification for G22-Davis.

Fig. 6d shows cloud phase distributions according to the G22-Christchurch cloud mask. The frequency of SLCC is highest

at 1.5 km and gradually decreases with altitude until around 7 km, following a very similar pattern to the VDR reference mask's SLCC occurrence. WLCC occurrence also closely follows the reference mask's WLCC occurrence, with frequencies highest at low altitudes, decreasing with altitude until around 2 km. As discussed earlier in this section, visual analysis of the G22-Christchurch cloud mask found that WLCC layers were occasionally misclassified as ICC. Fig. 6d confirms this is true throughout the dataset, shown by the large apparent peak in ICC frequency below 2 km. By comparison with the reference

mask in Fig. 6a, it is clear that this should be WLCC. In Fig. 6e, the $f_1$ score for SLCC detection as a function of altitude is relatively consistent and is higher than the $f_1$ score for G22-Davis in Fig. 6c, at all altitudes. Below 1 km there are no SLCC clouds present in either the VDR reference or the G22-Christchurch masks to allow comparison, so the F1 score drops to zero.

### 3.5   XGBoost feature analysis with SHAP

The previous sections have shown that our model performs very well at classifying backscatter peaks as SLCC or WLCC and

reasonably well at generating cloud masks of SLCC/WLCC occurrence which are comparable to the reference VDR cloud mask. In order to be confident the models can be applied in future work to ceilometer datasets from a range of locations, we need to understand the relative importance of the input features to the XGBoost algorithm (our 8-feature set of peak properties).

Fig. 7 shows correlation coefficients between each pair of peak properties, for all peaks in the dataset. This allows for a visual analysis of the independence of the XGBoost input features. Fig. 7 shows that peak height, width height and prominence are

strongly positively correlated, as we expect. Peak altitude and peak temperature are also strongly negatively correlated because low altitudes are associated with warmer temperatures, and high altitudes are associated with colder temperatures. We can also use Fig. 7 to select a subset of features that are independent by removing strongly correlated features. We use this information in Section 3.6 to train a modified model using a smaller range of independent input features.

Like Guyot et al. (2022), we apply the tree-based model explanation method by Lundberg et al. (2020) based on SHapley

Additive exPlanations (SHAP) values, which quantify the contributions to the model output from each feature. Figure 8 shows the SHAP value distribution for the input features to G22-Christchurch. In this stacked bar plot, features are ranked from top to bottom according to their mean absolute SHAP value, i.e. from most to least important for the model's classification. The distribution of SHAP values for the three most important features (peak width, altitude and temperature) are further shown in Fig. 9, which show SHAP value scatter plots corresponding to each feature, for each class.

Temperature is shown in Fig. 8 to be the most important feature for SLCC and WLCC classification, and the relationships shown in Figs. 9c, f and i reveal how XGBoost uses temperature to improve classifications. Fig. 9f shows that SHAP values for



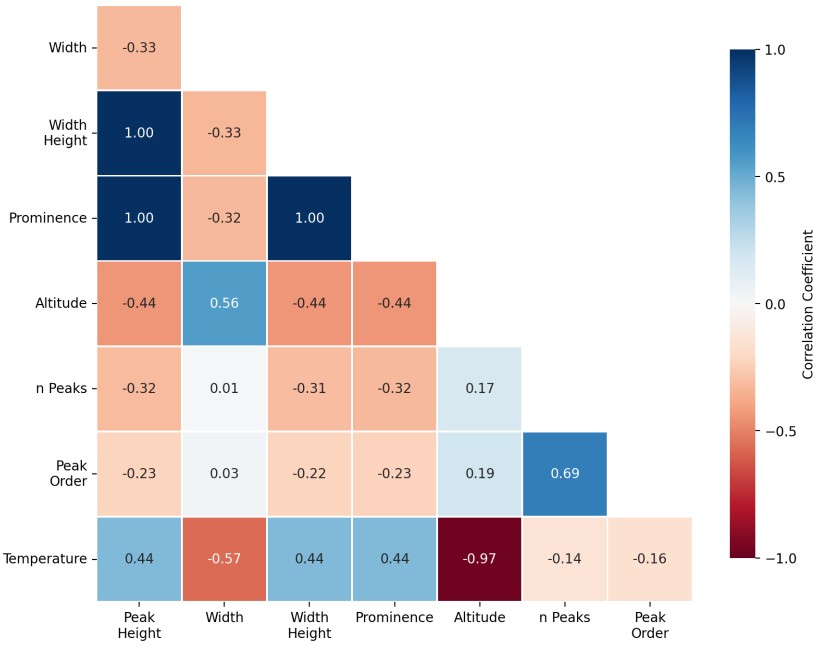

**Figure 7.** Correlation coefficients between each pair of peak properties, for all peaks in the dataset.

SLCC classification are positive for temperatures between -40 °C and 0 °C, and negative outside this range. On the other hand, the SHAP values for WLCC classification are positive above 0 °C and negative below 0 °C. The strong discontinuity at 0 °C for both SLCC and WLCC classification occurs because the VDR reference mask uses that temperature for distinguishing between
SLCC and WLCC. However, the strong discontinuity at around -40 °C is purely due to the distribution of peak temperatures in the training dataset, since no SLCC peaks were found below this temperature. This agrees with our physical understanding that SLW can exist at temperatures as low as around -40 °C, below which homogeneous freezing occurs (DeMott and Rogers, 1990). Fig. 9c shows that the opposite is true for ICC classification: SHAP values are strongly positive below around -40 °C (since only ice is found at these temperatures), and sometimes slightly positive for higher temperatures.

Altitude and temperature are inversely correlated (as shown in Fig. 7) because low altitudes are associated with warmer temperatures, and high altitudes are associated with colder temperatures. Fig. 8 shows peak altitude to be the second-most useful feature after temperature, with mean absolute SHAP values roughly equal across all classes. Fig. 9b shows that higher values of altitude were more strongly associated with ICC, and that low altitudes had a negative impact on ICC classification. The SHAP value distribution for SLCC classification (Fig. 9e) shows that SHAP values decrease with altitude above around
1 km, due to decreasing SLCC frequency with altitude in the training dataset. Below 1 km, a strong discontinuity means low altitudes have a negative impact on SLCC classification, as expected (since SLCC was rarely found at low altitudes). Fig. 9h shows the inverse is true for WLCC: low values of altitude with positive SHAP values are associated with a positive impact





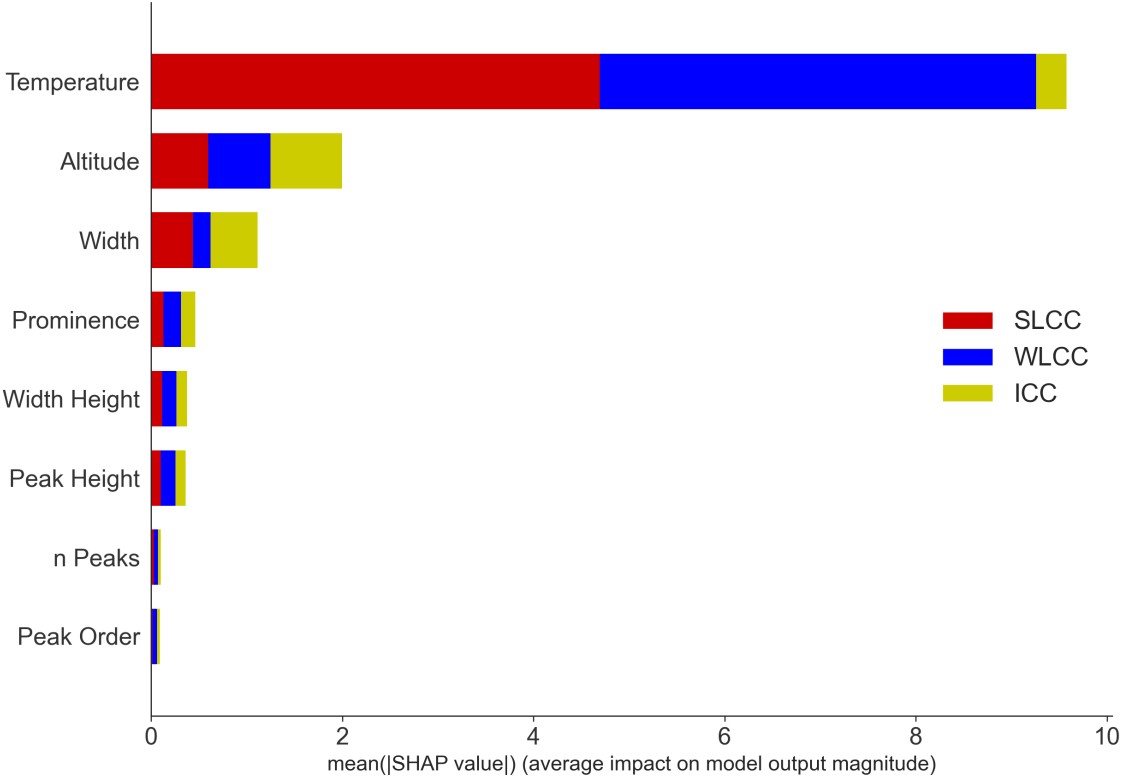

**Figure 8.** Mean absolute SHAP values for G22-Christchurch features, for each class (SLCC, WLCC and ICC). Features are ranked from top to bottom according to the sum of the mean absolute SHAP value across all classes, i.e. most to least important for the overall classification.

on WLCC classification, and these SHAP values decrease with altitude. As altitude increases, SHAP values become more negative, meaning these values have a negative impact on WLCC classification.

Next, Fig. 8 shows peak width to be a highly useful feature. The SHAP value scatter plots shown in Fig. 9a and d show the relationship between peak width and ICC/SLCC classification. Fig. 9d shows that low values of peak width had a positive impact on SLCC classification, and that high values had a negative impact. This agrees with our physical understanding of the properties of backscatter from LCC as discussed previously, namely that liquid water rapidly attenuates the lidar signal leading to a narrow band of enhanced returned backscatter. The reverse is true for ICC classification, shown in Fig. 9a, here a low value

of peak width is associated with a negative ICC classification, and higher values of peak width had a positive impact on ICC classification.

While Fig. 2 shows that there was a slight separation in the distribution of LCC and ICC peak backscatter heights, the SHAP values in Fig. 8 show that peak height was not a highly useful feature for XGBoost, in any class. Peak prominence, which we show in Fig. 7 is strongly positively correlated with peak height, has a higher mean absolute SHAP value, indicating it is more





**Figure 9.** SHAP value scatter plots for peak width, altitude and temperature for ICC (a-c), SLCC (d-f) and WLCC (g-i).



useful than peak height for classification. Scatter plots of SHAP values found that low prominence peaks had a positive impact
on ICC classification (not shown). One possibility for why peak height is less useful than peak prominence is that peak height
is already the main criteria used to select peaks. That is, we only analyse peaks that already exceed a backscatter threshold
of $\beta > 2 \times 10^{-5} \mathrm{m}^{-1}\mathrm{sr}^{-1}$, which excludes most low-$\beta$ peaks associated with ICC. Peak prominence, however, provides a way
of separating peaks with a higher signal to noise ratio (SNR), and is therefore more useful than peak height. The results here

differ from the results found by Guyot et al. (2022). In that study, peak height was found to be the most significant feature
used by XGBoost for the classification of SLCC. However, the aim of the G22-Davis model was to distinguish SLCC and
ice, while our aim is to distinguish SLCC, WLCC and ice. Therefore, it makes sense that our highest-scoring features are
temperature and altitude, which are powerful at distinguishing SLCC and WLCC (as shown in Fig. 3); whereas Guyot et al.'s
highest-scoring features were peak height, which was powerful at distinguishing SLCC and ice. Therefore, the SHAP value

differences between G22-Davis and G22-Christchurch are likely a consequence of the different classification objectives of the
two models.

## 3.6 Reducing XGBoost feature dimensionality

Due to the strong correlation between input features, as shown in Fig. 7, we next investigate the effect of reducing the number of
input features to train a modified XGB model. Reducing the dimensionality of the training dataset has been shown to improve

model performance in general by reducing overfitting (Russell, 2010). Furthermore, excluding temperature would remove the
dependence on AMPS or other numerical weather prediction model inputs which clearly include significant uncertainty. We
trained a new set of models using a reduced set of input features, testing various combinations of those input features with
the highest mean absolute SHAP values, as detailed in Fig. 8, and removing 'duplicate' features that are strongly correlated,
as determined in Fig. 7. For example, because peak height, prominence and width height are strongly correlated, we only use

one of those features as input to a model. The same is true for peak altitude and temperature, which are strongly negatively
correlated. We also remove the number of peaks and peak order because they have low absolute SHAP values, as shown in
Fig. 8. Table 1 shows the various input features and the balanced accuracy scores of the corresponding XGBoost models,
again tested with 3-fold group stratified cross-validation in each case. No hyperparameter tuning was completed during the
development of these models to allow a direct comparison with the original 8-feature model.

We see from Table 1 that the model with the best performing set of reduced input features ($T$, $w$, $\beta_{\mathrm{prom}}$) performed equally
well as the original model that used the complete set of 8 features. This model used peak temperature, peak width and peak
prominence, which we had previously shown using SHAP values to be among the most useful features. Replacing peak promi-
nence with either peak height or peak width height slightly reduced the accuracy, and removing it entirely (without replacement)
also only slightly reduced the accuracy. In fact, the balanced accuracy and SLCC $f_1$ score for the ($T$, $w$) model was slightly

higher than those scores for the ($T$, $w$, $\beta_w$) model, indicating that $\beta_w$ on average made the classification worse (although
this is not a significant difference). The model using just peak temperature as an input still gave reasonable accuracy scores,
although the drop of 0.09 in the SLCC $f_1$ score from the ($T$, $w$) model shows the importance of peak width as an input fea-
ture. It is worth noting, of course, that peak width and height are both essential features for identifying clouds and therefore





| | Balanced Accuracy | $f_1$ | | |
| --- | --- | --- | --- | --- |
| | | ICC | SLCC | WLCC |
| All Features | $0.87 \pm 0.02$ | $0.79 \pm 0.04$ | $0.85 \pm 0.02$ | $0.980 \pm 0.003$ |
| $T,\ w,\ \beta$ | $0.86 \pm 0.02$ | $0.77 \pm 0.04$ | $0.83 \pm 0.02$ | $0.980 \pm 0.003$ |
| $T,\ w,\ \beta_{\mathrm{prom}}$ | $0.87 \pm 0.02$ | $0.79 \pm 0.04$ | $0.85 \pm 0.02$ | $0.980 \pm 0.003$ |
| $T,\ w,\ \beta_w$ | $0.85 \pm 0.02$ | $0.76 \pm 0.04$ | $0.82 \pm 0.02$ | $0.980 \pm 0.003$ |
| $T,\ w$ | $0.86 \pm 0.02$ | $0.75 \pm 0.04$ | $0.83 \pm 0.02$ | $0.980 \pm 0.003$ |
| $T$ | $0.80 \pm 0.02$ | $0.69 \pm 0.04$ | $0.74 \pm 0.03$ | $0.980 \pm 0.004$ |
| $z,\ w,\ \beta$ | $0.78 \pm 0.007$ | $0.76 \pm 0.03$ | $0.66 \pm 0.1$ | $0.915 \pm 0.002$ |
| $z,\ w,\ \beta_{\mathrm{prom}}$ | $0.791 \pm 0.005$ | $0.79 \pm 0.03$ | $0.68 \pm 0.01$ | $0.915 \pm 0.002$ |
| $z,\ w,\ \beta_w$ | $0.77 \pm 0.01$ | $0.75 \pm 0.03$ | $0.65 \pm 0.01$ | $0.914 \pm 0.003$ |
| $z,\ w$ | $0.77 \pm 0.006$ | $0.75 \pm 0.04$ | $0.66 \pm 0.01$ | $0.915 \pm 0.005$ |
| $z$ | $0.72 \pm 0.01$ | $0.71 \pm 0.03$ | $0.54 \pm 0.01$ | $0.916 \pm 0.005$ |
| $w$ | $0.60 \pm 0.01$ | $0.69 \pm 0.03$ | $0.36 \pm 0.01$ | $0.80 \pm 0.01$ |

**Table 1.** Summary of the balanced accuracy scores and $f_1$ scores for each class from 3-fold group stratified cross-validation, for the adjusted models with reduced input features: peak temperature $T$, peak altitude $z$, peak width $w$, peak backscatter height $\beta$, peak prominence $\beta_{\mathrm{prom}}$ and peak width height $\beta_w$.

peaks in the first place. Replacing peak temperature with peak altitude reduced the balanced accuracy by around 0.08 for all models, and reduced the $f_1$ scores by 0.17 for SLCC and 0.07 for WLCC across all models. This significant reduction in model performance shows that despite peak altitude and peak temperature being strongly negatively correlated, altitude was not a useful direct replacement for temperature. This confirms the importance of having either NWP temperature information or radiosonde temperature data for making accurate classifications of SLCC. However, $f_1$ scores for WLCC were still high (>0.9) and unchanged for ICC (around 0.75) when temperature was replaced with altitude, indicating that LCC classification can generally be made without temperature information (i.e. without distinguishing SLCC).

## 4 Conclusions

In this study, we applied a method of supercooled liquid water containing cloud (SLCC) detection first introduced by Guyot et al. (2022) to observations from a mid-latitude site. From a 9-month dataset of MicroPulse Lidar (MPL) copolarized backscatter peak properties, we then trained an optimised gradient boosting model to classify backscatter peaks as SLCC, warm liquid water containing cloud (WLCC) or ice containing cloud (ICC). Unlike the binary SLCC classification model developed by Guyot et al. (2022), referred to as G22-Davis to reflect that the training dataset was from Davis Station, Antarctica, our model performed multi-class classification to distinguish SLCC from WLCC, which was common in our lidar observations from Christchurch, New Zealand.





We first used MPL depolarization observations to build a reference cloud phase mask which uses volume depolarization ratio (VDR) to separate ice and LCC, then uses Antarctic Mesoscale Prediction System (AMPS) air temperatures to distinguish SLCC and WLCC. Applying G22-Davis to our dataset of copolarized backscatter peak properties to create a SLCC classification mask, we obtained an overall balanced accuracy of 0.62, and balanced accuracy of 0.55 for days with detection of strong backscatter signals. We then trained and tested a modified XGBoost model, referred to as G22-Christchurch, and applied it to the full dataset to create a SLCC mask with overall balanced accuracy of 0.85, and balanced accuracy for days with detection of strong backscatter signals of 0.89. G22-Christchurch greatly improved classification of SLCC compared to G22-Davis, which often misclassified WLCC as SLCC due to the absence of warm liquid water in the Davis training data.

We also applied the tree-based model explanation method by Lundberg et al. (2020) based on SHapley Additive exPlanations (SHAP) values to quantify the relative importance of each XGBoost input feature (Lundberg and Lee, 2017). We found that temperature was the most important feature for SLCC and WLCC classification, due to the homogeneous freezing of liquid water at around -40 °C and the definition that SLCC exists below 0 °C. Peak width was the most important peak property for detecting liquid water of either type, because liquid water rapidly attenuates the lidar signal, causing a narrow peak in the returned backscatter profile. Peak prominence was also a useful feature for SLCC classification. We then developed a set of models with reduced input features, and compared their accuracies with the original model. We found that using only peak temperature, width and prominence as inputs, an XGBoost model could perform equally well as the original model trained using the full set of peak properties. Despite being strongly negatively correlated with temperature, peak altitude was not a suitable replacement for temperature for SLCC classification. This confirms the importance of air temperature data availability, such as from NWP models, for accurate detection of SLCC alongside ceilometer observations. This differs from the findings from Guyot et al. (2022), however it is important to note that our objectives differ. In this study, features that distinguish SLCC and WLCC are highlighted (i.e. temperature), whereas for Guyot et al. (2022), the best features that distinguished SLCC and ice were identified (i.e. peak height). When using our model to distinguish ice from LCC (of either type), the set of peak properties without temperature still gave good results, showing that LCC classification can be performed without relying on other sources for temperature information.

The frequency of SLCC occurrence was analysed for G22-Davis and G22-Christchurch and compared to our reference mask. Cloud fraction according to the reference mask was 14% for SLCC and 45% for WLCC. WLCC was only present at low altitudes, below 2 km, and SLCC was present between 1-8 km. G22-Davis often misclassified WLCC as SLCC which caused that model to incorrectly inflate SLCC occurrence below 2 km. On the other hand, G22-Christchurch replicated the reference mask's vertical structure of SLCC and WLCC occurrence, although WLCC occurrence was still underestimated. We hypothesise that this was due to misclassification of WLCC by the reference mask, and problems with the cloud mask generation technique.

The limitations of the relatively simple VDR reference mask should be noted. Precipitation and fog were not represented by the reference mask and may have been misidentified as ICC or WLCC. Previous methods (Tuononen et al., 2019; Guyot et al., 2022) have classified precipitation and fog using attenuated backscatter thresholds and gradients, which was not performed in this study. This may have artificially inflated the attenuated backscatter coefficient and therefore peak heights in WLCC and



ICC clouds. The effects of multiple scattering may also have caused some liquid layers to be misclassified as ICC. On the other
hand, horizontally aligned ice crystals likely caused some ice cloud to be misidentified as LCC, since the MPL was oriented
toward the zenith. This effect may have also caused ice particles to return higher than normal copolarized attenuated backscatter
(as described by Hogan and Illingworth, 2003), potentially influencing the distribution of ICC peak properties and therefore the
usefulness of peak height for G22-Christchurch in distinguishing LCC and ICC. The reference mask classification of SLCC
and WLCC relies on AMPS temperature data, which has its own associated uncertainty. In future work, more focus should be
given towards quantifying that error, such as using in-situ (e.g. radiosondes) or remote measurements of cloud temperature.

While G22-Christchurch was generally successful at reproducing the reference mask's classification for a peak, more work is
needed to accurately create a cloud mask product. The G22-Christchurch cloud mask was shown to underestimate the amount
of WLCC and overestimate the amount of ICC at low altitudes. While this can be partially attributed to misclassification of
WLCC by the model, it was also noted that the cloud mask generation technique was flawed. Every non-LCC cloudy bin
was labelled ICC, which was a problem when either no peaks were detected or when the cloud mask had not generated a
broad enough LCC layer to cover the surrounding ICC bins, as discussed in Section 3.4. The G22-Christchurch technique is
still useful, however, for identifying layers of LCC cloud from below and can be used to complement other ground-based or
satellite cloud masking products.

The aim of this study was to understand the limitations of G22-Davis in a mid-latitude site where both supercooled liquid and
warm liquid water clouds exist. Ceilometers, being relatively common and low-cost, have the potential to be a useful tool for
the detection of SLCC, and have particular value in regions where observations are sparse, such as Antarctica and the Southern
Ocean. Ground-based ceilometer observations, complementary to satellite observations, have the potential to improve cloud
phase products. We found that while the methodology of G22-Davis was successful when used with a local training dataset, the
original Davis-trained model was not effective at classifying SLCC over Christchurch. This confirms the need for further testing
in different regions under different environmental conditions. Future work could apply G22-Christchurch to other ceilometer
datasets from mid-latitude sites.

*Code and data availability.* The ALCF is open-source and available at https://alcf.peterkuma.net (last access: 19 May 2023) as well as a
permanent archivce of code and technical documentation on Zenodo at https://doi.org/10.5281/zenodo.5153867 (Kuma et al., 2021a, b). A
tool for converting MicroPulse Lidar data files to netCDF mpl2nc is open-source and available at https://doi.org/10.5281/zenodo.4409731
(Kuma, 2020). AMPS data was downloaded from https://www2.mmm.ucar.edu/rt/amps/ (last access: 19 May 2023). The observational data
(ALCF-processed netCDF MPL and cloud mask files) will be available on Zenodo by the end of 2023. The code and G22-Christchurch
algorithm described herein are available on request.

*Author contributions.* LW performed the code development, overall data analysis and prepared the manuscript with contributions from all
co-authors. AJM and AG provided regular scientific inputs on the analysis. LW and AG designed the experimental setup in Christchurch to
gather ground-based observations. AG provided the G22-Davis algorithm.



*Competing interests.* No competing interests identified.

*Acknowledgements.* AJM wishes to thank the Brian Mason Scientific and Technical Trust and the Deep South National Science Challenge for supporting the work presented and the scholarship.





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
