# Peer review of "Supercooled liquid water cloud classification using lidar backscatter peak properties"

_EGUsphere, 2023_

## Referee Comment (RC1)

Review of manuscript egusphere-2023-1085

**Title:** Supercooled liquid water cloud classification using lidar backscatter peak properties

**Authors:** Luke Whitehead, Adrian McDonald, and Adrien Guyot

Summary:

In this manuscript, Whitehead and coauthors examine a machine learning algorithm (an extreme gradient boosting or XGBoost model) trained to identify warm liquid clouds, supercooled clouds and ice containing clouds based on ceilometer observations and associated layer temperatures. The algorithm is trained against cloud phase determined using lidar volume depolarization ratio measurements, which traditional ceilometers do not measure.

Overview:

While I agree with the authors that a lidar/ceilometer-based technique that determines cloud (layer) phase without reliance on lidar depolarization would be useful, I think the current algorithm and analysis needs a bit more work.

Recommendation: Publish in with major revisions.

General Comments:

1) Too much ice cloud and too little water cloud

My largest concern is that the algorithm, as-is, clearly substantially overestimates the occurrence of ICC, while underestimating both SLCC and WLCC.

Looking at Figure 6, I would guess that 1/3 or so of the supercooled clouds near 1.5 km (where the occurrence is largest in the reference dataset) is missing. That is, the failed detection rate (see point #2 below) is significant. This doesn't make the algorithm useless by any means, but I think this needs to be quantified and summarized in the abstract and conclusions.

It also very striking the degree to which ICC is being overestimated, especially but not limited to, altitudes below 2 km. See below specific comments starting at lines 361 and 377. As described in the specific comments, I think the reasons for this problem need to be explored further and I suspect the situation can be improved rather easily.

2) False and Failed detection rates.

The analysis is largely focused on the f1 score and the "balanced accuracy" score which averages f1 across the phase categories (if I understand correctly). While this is OK, it

doesn't remove the value of knowing (and need to provide) the false and failed detection rates for each category, where:

False detection rate = (# of false positive detections / # of positive detections). This is the same as 1 – precision.

and

Failed detection rate = (# of failed detections / true total # of events in category). This is the same as 1 - recall.

I can't reconstruct these rates given only the f1 values (but I can get f1 score given only these two).

I don't mind if you use precision and recall rather than False and Failed detection rates (different fields somewhat different terminology) but at a minimum you need to provide at least a table (similar to Table 1) giving the recall and precision numbers and nominally plot the vertical profiles of recall and precision (as per Fig 6c and 6e) for the three phase categories.

3) A summary of the problems and limitations also needs to go into the abstract.

In my view, the abstract also needs to talk about the problems, presenting a more balanced view of the performance. The conclusion is reasonably well balanced but the abstract is not.

4) Multiple scattering and the VDR cloud phase

As discussed at several points in the manuscript, the VDR phase determination is far from perfect, and in particular multiple scattering can often increase the VDR and result in a miss identification of liquid phase cloud particles and ice phase. One can account for the multiple scattering (at least for optically thicker clouds). See for example, Mace et al (2020). Ideally, I would like to have seen an approach such as Mace's used as the reference, but such is obviously a major task and perhaps beyond the scope of what can be done with regard to the present manuscript. But I would encourage such as part of any continued development.

Mace, G. G., Benson, S., & Hu, Y. (2020). On the frequency of occurrence of the ice phase in supercooled Southern Ocean low clouds derived from CALIPSO and CloudSat. *Geophysical Research Letters*, *47*, e2020GL087554. https://doi.org/10.1029/2020GL087554

Specific Comments:

Line 29. Perhaps change " .. satellite-based measurements of …" to "satellite-based identification …".

Line 29. Perhaps change SLW to "SLW and Mixed Phase".

Line 34. "Transmission" not "emission". Emission is the process of radiating, transmission is the process of transmitting.

Line 36. I have never heard the term "Automatic lidar" or the "ALC" acronym before this manuscript. What is a non-automatic lidar? As far as I can see this acronym is only used one other time (on the next line 38/37). Perhaps remove entirely from the manuscript and write simply, "In this study we use a Micropulse lidar which measures the linear …"

*Line 117-121. Why did you subsample the VDR rather than average? This is especially confusing to me since in the next set of lines (119 to 121) you appear to be averaging these sub-samples to reduce noise??

Line 119-121. Perhaps expand the description of the processing here. As is, it appears that (1) you averaged the ratio rather than averaging the individual parallel and perpendicular backscattering components that go into Equation 1. The latter (averaging backscatter components not the ratio) is more physically sound as one expects noise to affect the ratio in a non-linear way that can amplify the effect of the noise (When noise makes $P_\parallel$ small it amplifies error). (2) Did you threshold before averaging? Again, this is not typically a good idea as it will typically generate bias.

Line 126. Please expand on this comment. I think of mixed phase clouds as being a combination of small (cloud-sized) liquid droplets and large (precipitation-sized) ice particles. I presume you mean that within the cloud, the scattering is dominated by cloud droplets and so one can't easily determine if precipitation sized particles are present, nor determine their phase. However, one can determine the phase of precipitation that is falling below cloud base, and in this way, identified mixed phase clouds which contain supercooled cloud droplets and precipitation ice (below).

Line 128 & 374. " … since ice cannot exist above 0 ◦C." Well, it takes time for ice to melt and it is quite common to have precipitating ice above 0 C. I think it is fair to say that small cloud-ice does typically exist below 0 ◦C. Perhaps rephrase to be more technically correct.

Line 140. The term "peak height" is potentially confusing as one might take height to mean altitude rather than the strength of the backscatter. Why not refer to this as the "peak magnitude"?

*Line 141. "Peak width height" is also potentially quite confusing. Further the meaning of "baseline" is not clear to me since one expect the backscatter not have equal values on both sides of the peak. An illustration of all eight characteristic would be very helpful here!

Line 186. Are the 5-minute time samples (I think this is what you are using in your statistics) within the same day being partitioned into different folds (both training and test folds)? Or is data for entire days (chunks of data) going into one fold or another? If the former, I am not sure it makes sense to talk about the data being independent. (Don't get me wrong it is good that you keep test data separate from the training data, but the 5 minute data are going to be highly correlated, and ideally you would partition data in time-chunks that are large enough that temporal correlation between the chunks is small).

Figure 4b. This is not showing all VDR values, just those where peaks are present, yes? This is not a problem per se, but perhaps worth explaining. In my view it would be nice to see the full VDR field (to get a sense for multiple scattering).

Line 197-201. What does this imply? Why not use the smallest / simplest network in this case (for if no other reason than to minimized the potential of overfitting)?

Line 299-310. I'm confused. I thought "balanced" meant averaged across phase categories (SLCC, WLCC, ICC) but here you are only describing the (binary) SLCC mask or WLCC masks?

Line 330. Perhaps worth discussing here is the low level clouds after 18 UTC which is water according to the VDR but ice to both of G22-Davis and G22-Christchurch. (Upon review, I see that you discuss this later in the manuscript. Perhaps just note here you discuss this cloud layer at a later point.)

Line 330. As best I understand, the miss-classification of the low cloud will have no or little effect on the accuracy statistics of SLCC presented here because the column contains SLW. If yes, perhaps point this out.

Line 332. On a very minor point but perhaps comment on how do you know this is a "thick band" of cloud. Perhaps simply note the lidar is being fully or heavily attenuated (and one is not seeing the top of cloud) here.

Line 340. I wrote "WHY?" in big letters in the margins on my first reading. Again, perhaps note that this problem is discussed in more detail later in the manuscript.

**Line 361-366 and 389-391. While I'm sure the VDR does misclassify some WLCC as ice because of multiple scattering, both of your examples and Figure 6a seems to suggest this is a very small percentage of the data. And if so, shouldn't the algorithm have learned that such warm clouds near the surface are rarely ice and therefore it should guess that warm clouds are liquid (rather than infrequently occurring ice)? I would wager that if you look carefully at the backscatter characteristics or masking/peak detection issues (see comment line 377) that these will have a lot more to do with this overestimate. When use later use ONLY temperature as an input, a discussed later in the manuscript, what happened to these low clouds? I require more evidence to believe that a small percentage of bad data in the reference set is really the cause of this problem.

Line 365.  You write "This demonstrates the sensitivity of the reference mask, and thus the G22-Christchurch model, to the VDR threshold δLCC, and suggests that δLCC should be increased."  I entirely disagree with this in part because of my comment above (Line 361).  In general, doing so will increase the amount of time that ice is being called liquid in the reference dataset.  Rather, I think a better solution would be use a reference technique that accounts for multiple scattering as part of the training (see general comment #4).

*Figure 6.  This is an important figure.  Panels (c) and (e) need more tick marks and need to be expanded (or something) so one can read values.  As per general comment #2, please also add recall/precision rates (and discuss such in the text).  In general, it would be good to plot values for WLCC and ICC (not just SLCC).

**Line 377.  You wrote "… other parts of the cloud are identified as ICC by the mask because there are either no peaks detected (so the cloudy bins in that profile are conservatively labelled ICC by default)."  So this seems like a large potential source of the ICC overestimation problem.   But if I understand, you know what regions are peaks (and neighboring width pixels) and which are not.  And if so, you should be able to easily establish the degree to which this is a problem.   And it occurs to me that simple potential fixes might be: (1) To either assume points not associated with a peak are liquid, if they are warm and ice otherwise.   I note here that clouds with top temperatures only a few degrees below zero (say warmer than about -5 C) tend to be liquid, so personally I would use cloud-top-temperature.  AND/or (2) assign the phase based on the closest peak within the same cloudy layer, or near the same altitude in a neighboring column.

*Line 395.  It looks to me that 1/3 or more of the SLCC might be missing.

Line 398.  "occasionally" misclassified?   I think Figure 6d shows that this is a lot more than just occasionally.   Much of the WLCC is missing.

Line 406.  You write "In order to be confident the models can be applied in future work to ceilometer datasets from a range of locations, …".  So I think being applicable to other locations is an excellent goal and something well worth testing. You leave this for future work in the conclusion and that is OK, but presumably you have the Davis station data from the Guyot (who is a co-author) and the same VDR reference data so it would be easy to at least see what happens when you apply the G22-Chirstchurch algorithm to the Davis data.

Line 487.   I like your analysis in this section (though I would like to see a table for recall and precision – is the story the same for these metrics)?  Perhaps it is beyond the scope of what you are willing to do here but using surface temperature or surface temperature and altitude might also work well and thereby avoid the dependence on radiosondes or NWP.

Line 489-90.  This is a good point.  But it seems to me that it is an open question as to whether the relationship between temperature and SLCC occurrence is constant (doesn't vary with location).  In general, to me, you results argue for using the <T, omega> or <T,

omega, Beta_prom> version of the code as your "default algorithm" (that I hope you would provided to other researchers).

Line 539-40.   See general comment #4.

---

## Author Comment (AC1)

egusphere-2023-1085

Supercooled liquid water cloud classification using lidar backscatter peak properties

**Authors' response to referee comments**

We would like to thank the two anonymous referees for their valuable comments which we believe have improved the quality of the work and the manuscript. This work also received comments from an internal reviewer at the Australian Bureau of Meteorology, who we also thank for their valuable input.

The main changes are summarised here:
1) The most significant change in the revised manuscript is the new procedure and terminology for the reference cloud mask classification. We now use the terms supercooled liquid water cloud (SLWC), supercooled liquid water containing cloud (SLCC), mixed phase cloud (MPC), warm liquid water cloud (WLWC) and ice cloud (IC). The new reference mask classification is discussed in detail in Section 2.2.1 in the revised manuscript. Liquid cloud, MPC and IC are distinguished according to VDR thresholds of 0.1 and 0.4, which were determined following a sensitivity test on the values of VDR for all cloudy bins in our MPL dataset (shown in the new Figure 1). For liquid clouds, SLWC and WLWC are then distinguished using WRF air temperatures. The updated reference mask is described in detail in the revised manuscript in lines 120-136.
2) The XGBoost algorithm was retrained against the new reference mask, and now performs multiclass classification of backscatter peaks as SLWC, WLWC or neither (i.e. MPC/IC). Model setup was changed to include further hyperparameter testing to improve performance and avoid overfitting.
3) While model performance is discussed in Section 3.1, performance metrics relating to SLWC are the focus since the goal of this study is to use ceilometers to detect SLWC occurrence, not to provide a comprehensive cloud phase categorization mask. Given the implicit uncertainty in using a lidar-only reference mask to estimate MPC occurrence, we do not focus on the performance of MPC classification.

A tracked changes document (made with latexdiff) is attached along with the revised manuscript.

Reviewer 1

Summary:
In this manuscript, Whitehead and coauthors examine a machine learning algorithm (an extreme gradient boosting or XGBoost model) trained to identify warm liquid clouds, supercooled clouds and ice containing clouds based on ceilometer observations and associated layer temperatures. The algorithm is trained against cloud phase determined using lidar volume depolarization ratio measurements, which traditional ceilometers do not measure.

Overview:
While I agree with the authors that a lidar/ceilometer-based technique that determines cloud (layer) phase without reliance on lidar depolarization would be useful, I think the current algorithm and analysis needs a bit more work.

Recommendation: Publish in with major revisions.

General Comments:

1) Too much ice cloud and too little water cloud
My largest concern is that the algorithm, as-is, clearly substantially overestimates the occurrence of ICC, while underestimating both SLCC and WLCC.
Looking at Figure 6, I would guess that 1/3 or so of the supercooled clouds near 1.5 km (where the occurrence is largest in the reference dataset) is missing. That is, the failed detection rate (see point #2 below) is significant. This doesn't make the algorithm useless by any means, but I think this needs to be quantified and summarized in the abstract and conclusions. It also very striking the degree to which ICC is being overestimated, especially but not limited to, altitudes below 2 km. See below specific comments starting at lines 361 and 377. As described in the specific comments, I think the reasons for this problem need to be explored further and I suspect the situation can be improved rather easily.

R1C1: Following the implementation of the improved reference mask classification and further model training, we found the model performance improved, and the updated figures shown in Figs. 5 and 6 demonstrate these improvements. The specific comments below discuss these changes in more detail.

2) False and Failed detection rates.
The analysis is largely focused on the f1 score and the "balanced accuracy" score which averages f1 across the phase categories (if I understand correctly). While this is OK, it doesn't remove the value of knowing (and need to provide) the false and failed detection rates for each category, where: False detection rate = (# of false positive detections / # of positive detections). This is the same as 1 – precision. And Failed detection rate = (# of failed detections / true total # of events in category). This is the same as 1 – recall. I can't reconstruct these rates given only the f1 values (but I can get f1 score given only these two). I don't mind if you use precision and recall rather than False and Failed detection rates (different fields somewhat different terminology) but at a minimum you need to provide at least a table (similar to Table 1) giving the recall and precision numbers and nominally plot the vertical profiles of recall and precision (as per Fig 6c and 6e) for the three phase categories.

R1C2: Thanks for this suggestion. These changes were implemented and now recall and precision are the main metrics used when discussing the accuracy of each class (i.e. the ability of the model to detect SLWC, or WLWC, or IC etc.). The f1 score, defined as the harmonic mean of precision and recall for a given class, can be reconstructed from precision and recall if required. The balanced accuracy, defined as the average of the recalls on each class, is used more generally to describe the overall performance of the model. Fig 7 (formerly Fig 6) was updated to show the precision and recall of the SLWC class as a function of altitude. Table 4 (formerly Table 1) was updated to show precision and recall scores.

3) A summary of the problems and limitations also needs to go into the abstract. In my view, the abstract also needs to talk about the problems, presenting a more balanced view of the performance. The conclusion is reasonably well balanced but the abstract is not.

R1C3: The following line in the abstract was updated:

"The performance of our new model, trained using data from Christchurch, displays recall scores as high as 0.88 for identification of SLWC, although generally underestimates SLWC occurrence."

4) Multiple scattering and the VDR cloud phase
As discussed at several points in the manuscript, the VDR phase determination is far from perfect, and in particular multiple scattering can often increase the VDR and result in a miss identification of liquid phase cloud particles and ice phase. One can account for the multiple scattering (at least for optically thicker clouds). See for example, Mace et al (2020). Ideally, I would like to have seen an approach such as Mace's used as the reference, but such is obviously a major task and perhaps beyond the scope of what can be done with regard to the present manuscript. But I would encourage such as part of any continued development.
Mace, G. G., Benson, S., & Hu, Y. (2020). On the frequency of occurrence of the ice phase in supercooled Southern Ocean low clouds derived from CALIPSO and CloudSat. Geophysical Research Letters, 47, e2020GL087554. https://doi.org/10.1029/2020GL087554

R1C4: The reference mask was updated, as discussed in the first main comment above. Figure 1 demonstrates that the new VDR classification matches well with the pattern of change in VDR related to temperature. We therefore feel that this strengthens our result, though we acknowledge that the VDR mask is not perfect, as discussed in Section 2.2.

Specific Comments:
Line 29. Perhaps change " .. satellite-based measurements of …" to "satellite-based identification …".

R1C5: Corrected.

Line 29. Perhaps change SLW to "SLW and Mixed Phase".

R1C6: We now refer to supercooled liquid water containing cloud (SLCC), which includes SLW and mixed phase as defined in line 18.

Line 34. "Transmission" not "emission". Emission is the process of radiating,

transmission is the process of transmitting.

R1C7: Corrected.

Line 36. I have never heard the term "Automatic lidar" or the "ALC" acronym before this manuscript. What is a non-automatic lidar? As far as I can see this acronym is only used one other time (on the next line 38/37). Perhaps remove entirely from the manuscript and write simply, "In this study we use a Micropulse lidar which measures the linear …"

R1C8: The term 'Automatic lidar' has been used in previous studies to describe relatively low-cost, low-maintenance single-wavelength elastic backscatter lidars, as opposed to more complex multi-wavelength lidars (for example Kuma et al., 2021; Madonna et al., 2018; Dionisi et al., 2018; Kotthaus et al. 2018). However, to avoid confusion, we removed the usage of the term and followed the reviewer's suggestion. The line now reads: "In this study we use a MicroPulse Lidar (MPL), which has a depolarization capability..."

*Line 117-121. Why did you subsample the VDR rather than average? This is especially confusing to me since in the next set of lines (119 to 121) you appear to be averaging these sub-samples to reduce noise??

R1C9: Agreed that the original wording was unclear. The lidar data was actually regridded by averaging to the lower resolution of the ALCF-processed grid (5 minute, 50 m resolution), which removes noise at the expense of higher resolution.

Line 119-121. Perhaps expand the description of the processing here. As is, it appears that (1) you averaged the ratio rather than averaging the individual parallel and perpendicular backscattering components that go into Equation 1. The latter (averaging backscatter components not the ratio) is more physically sound as one expects noise to affect the ratio in a non-linear way that can amplify the effect of the noise (When noise makes $P_\parallel$ small it amplifies error). (2) Did you threshold before averaging? Again, this is not typically a good idea as it will typically generate bias.

R1C10: In the updated reference mask processing, the individual parallel and perpendicular backscatter components are regridded by averaging before the depolarization ratio is calculated. We agree that this is more physically sound. The updated processing method now classifies cloud bins according to the cloud phase diagnostic criteria in Table 1, following a similar method for processing MPL measurements as Lewis et al. (2020). This is described in the revised manuscript in lines 120-136.

Line 126. Please expand on this comment. I think of mixed phase clouds as being a combination of small (cloud-sized) liquid droplets and large (precipitation-sized) ice particles. I presume you mean that within the cloud, the scattering is dominated by cloud droplets and so one can't easily determine if precipitation sized particles are present, nor determine their phase. However, one can determine the phase of precipitation that is falling below cloud base, and in this way, identified mixed phase clouds which contain supercooled cloud droplets and precipitation ice (below).

R1C11: In the updated reference mask, SLW and MPC are distinguished based on VDR. See R1C4. Therefore, the line "SLW and mixed-phased clouds cannot be distinguished using this method" in the original manuscript at line 127 was removed.

Line 128 & 374. " … since ice cannot exist above 0 ∘C." Well, it takes time for ice to melt and it is quite common to have precipitating ice above 0 C. I think it is fair to say that small cloud-ice does typically exist below 0 ∘C. Perhaps rephrase to be more technically correct.

R1C12: It is correct that ice can exist at temperatures above 0°C before the onset of melting. This section was rephrased and the line " … since ice cannot exist above 0°C" was removed. See also R2C12.

Line 140. The term "peak height" is potentially confusing as one might take height to mean altitude rather than the strength of the backscatter. Why not refer to this as the "peak magnitude"?

R1C13: We agree that this was confusing, and now use the term peak magnitude throughout the revised manuscript to describe this property.

*Line 141. "Peak width height" is also potentially quite confusing. Further the meaning of "baseline" is not clear to me since one expect the backscatter not have equal values on both sides of the peak. An illustration of all eight characteristic would be very helpful here!

R1C14: This section was reworded to provide more clarity in the revised manuscript at lines 144-154. Figure 6 of Guyot et al. (2022) illustrates the backscatter peak features, and this is referenced in the revised text.

Line 186. Are the 5-minute time samples (I think this is what you are using in your statistics) within the same day being partitioned into different folds (both training and test folds)? Or is data for entire days (chunks of data) going into one fold or another? If the former, I am not sure it makes sense to talk about the data being independent. (Don't get me wrong it is good that you keep test data separate from the training data, but the 5 minute data are going to be highly correlated, and ideally you would partition data in time-chunks that are large enough that temporal correlation between the chunks is small).

R1C15: The groups for stratified 3-fold cross validation were partitioned by month, so nearby data are kept in the same train/test folds. This section was reworded for clarity, and the following line was added at line 197 in the revised manuscript:

"This means that highly-correlated neighbouring measurements are kept together in the same fold, ensuring each train/test fold is independent."

Figure 4b. This is not showing all VDR values, just those where peaks are present, yes? This is not a problem per se, but perhaps worth explaining. In my view it would be nice to see the full VDR field (to get a sense for multiple scattering).

R1C16: In the revised manuscript, Figure 5b/6b now show the full VDR field as suggested.

Line 197-201. What does this imply? Why not use the smallest / simplest network in this case (for if no other reason than to minimized the potential of overfitting)?

R1C17: The hyperparameters described on these lines are just two examples of many different possible hyperparameters to adjust. Reducing one does not necessarily guarantee simplicity and prevent overfitting. Instead, a grid search was performed with 3-fold group stratified cross-validation to find the best-performing hyperparameter combination, scoring by balanced accuracy. These best-performing hyperparameters were used in the model applied throughout the rest of the analysis. This section was reworded for clarity in the revised manuscript at lines 200-206.

Line 299-310. I'm confused. I thought "balanced" meant averaged across phase categories (SLCC, WLCC, ICC) but here you are only describing the (binary) SLCC mask or WLCC masks?

R1C18: In the revised manuscript, we now report only recall and precision scores in this section, for clarity.

Line 330. Perhaps worth discussing here is the low level clouds after 18 UTC which is water according to the VDR but ice to both of G22-Davis and G22-Christchurch. (Upon review, I see that you discuss this later in the manuscript. Perhaps just note here you discuss this cloud layer at a later point.)

R1C19: Following the reprocessing during the revisions, this layer of cloud is now accurately represented as WLWC by G22-Christchurch. The G22 cloud masks had previously called any 'Unknown' cloud IC, which led to inaccuracies like the one discussed here. This was addressed and improved upon in the updated version.

Line 330. As best I understand, the miss-classification of the low cloud will have no or little effect on the accuracy statistics of SLCC presented here because the column contains SLW. If yes, perhaps point this out.

R1C20: Again, in this case the low cloud is no longer misclassified.

Line 332. On a very minor point but perhaps comment on how do you know this is a "thick band" of cloud. Perhaps simply note the lidar is being fully or heavily attenuated (and one is not seeing the top of cloud) here.

R1C21: In the revised manuscript, this has been changed to "a wide band" for clarity.

Line 340. I wrote "WHY?" in big letters in the margins on my first reading. Again, perhaps note that this problem is discussed in more detail later in the manuscript.

R1C22: Again, in this case the low cloud is no longer misclassified.

**Line 361-366 and 389-391. While I'm sure the VDR does misclassify some WLCC as ice because of multiple scattering, both of your examples and Figure 6a seems to suggest this is a very small percentage of the data. And if so, shouldn't the algorithm have learned that such warm clouds near the surface are rarely ice and therefore it should guess

that warm clouds are liquid (rather than infrequently occurring ice)? I would wager that if you look carefully at the backscatter characteristics or masking/peak detection issues (see comment line 377) that these will have a lot more to do with this overestimate. When use later use ONLY temperature as an input, a discussed later in the manuscript, what happened to these low clouds? I require more evidence to believe that a small percentage of bad data in the reference set is really the cause of this problem.

R1C23: Following reprocessing and further investigation, this section of the original manuscript was removed. The new reference mask now does not detect IC for temperatures above 0°C, as shown in Figure 7 (formerly Fig. 6), instead labelling such cloud 'Unknown' (when VDR is also greater than 0.1). The new version of the model mask correctly labels this cloud as WLWC when a peak is detected, and Unknown when a peak is not detected.

Line 365. You write "This demonstrates the sensitivity of the reference mask, and thus the G22-Christchurch model, to the VDR threshold δLCC, and suggests that δLCC should be increased." I entirely disagree with this in part because of my comment above (Line 361). In general, doing so will increase the amount of time that ice is being called liquid in the reference dataset. Rather, I think a better solution would be use a reference technique that accounts for multiple scattering as part of the training (see general comment #4).

R1C24: We agree, and these problems are not present after using the new reference mask and model. This line is removed in the revised manuscript.

*Figure 6. This is an important figure. Panels (c) and (e) need more tick marks and need to be expanded (or something) so one can read values. As per general comment #2, please also add recall/precision rates (and discuss such in the text). In general, it would be good to plot values for WLCC and ICC (not just SLCC).

R1C25: Figure 6 (now Fig. 7) has been updated with these suggestions, with expanded scoring panels (c, e) showing recall and precision for the SLWC classification. Since detection of SLWC is the goal of this study, only accuracy statistics relating to SLWC are presented here. As mentioned in R1C1, the altitude statistics are not reliable for quantitatively evaluating model performance since they are sensitive to how the cloud mask vertical extent/thickness is defined. The following line was added to address this at line 373:

"We also found that the frequencies and accuracy scores in Figs. 7d, e were sensitive to the width of the generated cloud mask layer as defined by the upper and lower bounds around the peak location. Therefore, the cloud phase distributions in Fig. 7 provide qualitative evidence of the model's performance, but quantitative results are best determined by comparing time-step SLWC Booleans as in Section 3.2."

**Line 377. You wrote "… other parts of the cloud are identified as ICC by the mask because there are either no peaks detected (so the cloudy bins in that profile are conservatively labelled ICC by default)." So this seems like a large potential source of the ICC overestimation problem. But if I understand, you know what regions are peaks (and neighboring width pixels) and which are not. And if so, you should be able to easily establish the degree to which this is a problem. And it occurs to me that simple potential fixes might be: (1) To either assume points not associated with a peak are liquid, if they

are warm and ice otherwise. I note here that clouds with top temperatures only a few degrees below zero (say warmer than about -5 C) tend to be liquid, so personally I would use cloud-top-temperature. AND/or (2) assign the phase based on the closest peak within the same cloudy layer, or near the same altitude in a neighboring column.

R1C26: We agree, and this is the basis for the new model cloud mask, which instead labels clouds as 'Unknown' when backscatter is not strong enough for a peak to be detected.

*Line 395. It looks to me that 1/3 or more of the SLCC might be missing.

R1C27: In the reprocessed results, SLWC detection improved. As stated previously in R1C1 and R1C25, Fig 7 provides qualitative evidence of the model's performance, but quantitative results are best determined by comparing time-step SLWC Booleans as in Section 3.2

Line 398. "occasionally" misclassified? I think Figure 6d shows that this is a lot more than just occasionally. Much of the WLCC is missing.

R1C28: The new reprocessed results in Fig 7 show that the WLWC classification at low levels has improved.

Line 406. You write "In order to be confident the models can be applied in future work to ceilometer datasets from a range of locations, …". So I think being applicable to other locations is an excellent goal and something well worth testing. You leave this for future work in the conclusion and that is OK, but presumably you have the Davis station data from the Guyot (who is a co-author) and the same VDR reference data so it would be easy to at least see what happens when you apply the G22-Chirstchurch algorithm to the Davis data.

R1C29: Applying the G22-Christchurch algorithm to other sites and datasets is an important area for future work, but beyond the scope of this study.

Line 487. I like your analysis in this section (though I would like to see a table for recall and precision – is the story the same for these metrics)? Perhaps it is beyond the scope of what you are willing to do here but using surface temperature or surface temperature and altitude might also work well and thereby avoid the dependence on radiosondes or NWP.

R1C30: Table 4 (formerly Table 1) was updated to show recall/precision scores instead of f1 scores, as suggested.

Line 489-90. This is a good point. But it seems to me that it is an open question as to whether the relationship between temperature and SLCC occurrence is constant (doesn't vary with location). In general, to me, you results argue for using the <T, omega> or <T, omega, Beta_prom> version of the code as your "default algorithm" (that I hope you would provided to other researchers).

R1C31: Agreed, and added the following line at 501:

"For future work, such as the incorporation of this retrieval method in ALCF, the default XGBoost model will only use peak width, prominence and temperature as inputs, while a model using width, prominence and altitude as inputs will be an alternative for cases when temperature data is unavailable."

Line 539-40. See general comment #4.

R1C32: See R1C4

---

## Author Comment (AC2)

egusphere-2023-1085

**Supercooled liquid water cloud classification using lidar backscatter peak properties**

**Authors' response to referee comments**

We would like to thank the two anonymous referees for their valuable comments which we believe have improved the quality of the work and the manuscript. This work also received comments from an internal reviewer at the Australian Bureau of Meteorology, who we also thank for their valuable input.

The main changes are summarised here:

1) The most significant change in the revised manuscript is the new procedure and terminology for the reference cloud mask classification. We now use the terms supercooled liquid water cloud (SLWC), supercooled liquid water containing cloud (SLCC), mixed phase cloud (MPC), warm liquid water cloud (WLWC) and ice cloud (IC). The new reference mask classification is discussed in detail in Section 2.2.1 in the revised manuscript. Liquid cloud, MPC and IC are distinguished according to VDR thresholds of 0.1 and 0.4, which were determined following a sensitivity test on the values of VDR for all cloudy bins in our MPL dataset (shown in the new Figure 1). For liquid clouds, SLWC and WLWC are then distinguished using WRF air temperatures. The updated reference mask is described in detail in the revised manuscript in lines 120-136.

2) The XGBoost algorithm was retrained against the new reference mask, and now performs multiclass classification of backscatter peaks as SLWC, WLWC or neither (i.e. MPC/IC). Model setup was changed to include further hyperparameter testing to improve performance and avoid overfitting.

3) While model performance is discussed in Section 3.1, performance metrics relating to SLWC are the focus since the goal of this study is to use ceilometers to detect SLWC occurrence, not to provide a comprehensive cloud phase categorization mask. Given the implicit uncertainty in using a lidar-only reference mask to estimate MPC occurrence, we do not focus on the performance of MPC classification.

A tracked changes document (made with latexdiff) is attached along with the revised manuscript.

Reviewer comment 2

This review addresses the manuscript of Whitehead et al., 2023, entitled "Supercooled liquid water cloud classification using lidar backscatter peak properties". The study deals with the identification of layers of liquid water based on machine learning techniques applied to observations of a polarization-capable micro pulse lidar (MPL). While still not being super-familiar to the emerging world of machine learning, I was able to get a bit of more impressions on how AI might assist in future data analyses. Central instrument of the study is, besides a toolset of existing Python-based machine learning APIs, a polarimetric MPL lidar system which was operated at Christchurch, CA, NZ. Three retrievals were cross-evaluated against each other. The default one, provided by the MPL software, one ML retrieval for the Antarctic site of Davis, and a newly retrieved one for the site of Christchurch.

As far as I got, the default MPL retrieval was set as the reference dataset. It was found that the ML dataset for Davis had worse scoring compared to the retrieval based directly on observations at Christchurch.

Being allover well structured and well written, the study in the end appears somewhat inconclusive. To my impression, the conclusions drawn are not suited to provide guidance for future studies. I've learned that machine learning classifications cannot be transferred between different sites. I wonder if this is a good basis for any future (comparative) studies. How should valid scientific conclusions be drawn when the underlying datasets are individually tuned to single sites? What do the authors think about this issue? Isn't it more appropriate to use a physics-based retrieval which just requires a well-calibrated instrument? All further retrieval steps would then just be determined by the atmospheric state, without the inclusion of tuned ML-based decision trees.

Anyway – given the good structure and rather complete content, I consider the study as suited to be published in AMT. I nevertheless have a series of remarks and questions, which I would like the authors to reply on and to consider in the revised version of the manuscript. A second round of revision appears to me to be necessary in order to evaluate whether the identified issues/remarks could be accurately addressed.

Major comments:
I was missing an overview on the standard procedures of cloud detection. E.g., the widely used synergistic cloud retrievals such as ARSCL or Cloudnet use a combination of Att. BSC. threshold and gradient (https://doi.org/10.2172/1808567, https://joss.theoj.org/papers/10.21105/joss.02123 ... both just require the lidar for the liquid detection) and appear to be quite successful with this. In addition, both of these retrievals use different att. BSC thresholds which demonstrates that there's not the one and only solution. This information might be relevant to discuss, e.g., in line 101 of the manuscript.

R2C1: As per the reviewer's suggestion, more detail was added to Section 1 at line 58 to describe the earlier methods:

"Operational networks of comprehensive observing systems, such as the Atmospheric Radiation Measurement (ARM) Climate Research Facility (Mather and Voyles, 2013) and Cloudnet (Illingworth et al., 2007), use synergistic radar-lidar algorithms to retrieve cloud properties including cloud phase. Within the Cloudnet retrieval, liquid water detection is based on empirically-derived thresholds of lidar attenuated backscatter (Hogan et al., 2003; Illingworth et al., 2007) and in recent versions, the attenuated backscatter profile shape (Tuononen et al., 2019; Tukiainen et al., 2020). "

Note also that Guyot et al. (2022) compared the new machine-learning algorithm to the Cloudnet retrieval (Tuononen et al., 2019) and found the new technique outperformed the previous method, as described in Section 1.

Equation 1: This expression is only true for an ideal lidar systems with known system constants, the absence of any cross-talk between the channels and a 100% perfect polarization state of the emitted light. A general treatment of the depolarization calibration procedure is described by https://amt.copernicus.org/articles/9/4181/2016/.

R2C2: We use the same definition of depolarization ratio for cloud phase determination as has been used in previous studies (e.g. Guyot et al., 2022; Lewis et al., 2020).

Lines 148-166: Multi layers. I don't quite get what the offset is between two subsequent layers. I'd calculate the actual offset between Q2prime and Q2second 1.1e-4-4.4e-5, which yields 6.6e-5. Can the authors please clarify? I also don't understand the statement about the extinction in line 154. Extinction is not in units of m^-1 sr^-1.

R2C3: This section first reviews the preliminary analysis performed by Guyot et al. (2022) to compare the returned backscatter properties for primary peaks (first layer) and secondary peaks (all higher layers). In that study, their data showed that backscatter from primary peaks was higher than backscatter from secondary peaks, and that this was statistically significant (see Figure 3 from Guyot et al.). To ensure a fair comparison across primary and secondary peaks, the values of backscatter magnitude for secondary peaks were adjusted by adding an offset, which was calculated as the difference in the medians of the distributions. For our dataset, we compared the distributions of primary and secondary peaks magnitudes (shown in Figure 2) and found that they were not significantly different, so we chose not to adjust the secondary peaks.

The statement about extinction was rewritten at line 161:

"This was hypothesised to be the average reduction in lidar backscatter due to extinction from the lower layer(s)"

Minimum detection height: What is the minimum height for cloud layer detection? In Fig. 3e, it seems as if there is some certain gap between the ground and the first cloud bases. At least for the Arctic, low-level stratus clouds with a base below 100 m were recently highlighted to be a challenging but important puzzle piece in the Arctic cloud puzzle (Griesche et al., 2024; https://doi.org/10.5194/acp-24-597-2024). See the second case study in the manuscript, which indicates that there are issues near to the surface.

R2C4: The following line was added to Section 2.1.1 at line 86:

"The minimum range and detection height of the MPL is 100 m."

Furthermore, the following was added to the conclusion at lines 518:

 "Additionally, it should be noted that the minimum detection height of the MPL is 100 m. Therefore, the MPL analysis in this work potentially misses low-level cloud, which Griesche et al. (2024) identified as an important polar cloud regime for future observational studies."

3f: Do the authors have an explanation for the gap in SLCC and WLCC at 0°C? Looking at Fig. 2f, this gap is not so pronounced in the overall LCC distribution. This discussion could be added to lines 182-183.

R2C5: We don't have an explanation for why this gap appears, although it could appear due to a difference in the normalization of the SLWC and WLWC KDE plots in Figure 4f. In the revised manuscript, a comment on the mode (i.e. peak in the KDE plot) of each SLWC and WLWC distribution was added to lines 281-283, as suggested.

Case studies 1&2, Figs 4 and 5: Based on the case studies I get really puzzled. The performance of the G22 retrievals is visually just really low, especially for the low-level clouds in the evening of case study 2 (Fig. 5). On the other hand, the reference VDR retrieval misclassifies high aerosol loads as liquid cloud in case study 1 (Fig. 4). Intuitively, I would presume that both issues could be mitigated just by using a simple combination of threshold value and gradient. Why the machine learning?

R2C6: Following reprocessing of the reference mask and retraining of the G22-Christchurch model, SLWC detection improved.

Lines 476-477: After applying the SHAP filtering, three parameters remain as relevant: temperature, peak prominence (BSC threshold) and peak width (gradient). It's funny to see that the cleaned version of the G22 retrieval just condenses to the same parameters as are used in the standard threshold/gradient methods.

R2C7: We consider this a good sign – the SHAP analysis shows that the most useful features are physically interpretable, which increases our confidence in the XGBoost model.

Figure 6: Would be nice to have a version of frequency vs. temperature. This could help to evaluate the impact of specular reflection on the retrievals, as specular reflection (false liquid) should be most prominent in the temperature range between -10 and -20°C.

R2C8: KDE plots showing the temperature distributions of peaks (and therefore cloud frequency as a function of temperature) are already presented in Figures 3f for liquid and non-liquid, and Fig. 4f for SLWC and WLWC.

Final statement in the conclusion could be added: Which retrieval will the authors use in future for their studies? Can the authors make a decision and motivate it?

R2C9: We added the following two lines to the conclusion, at line 501:

"For future work, such as the incorporation of this retrieval method in ALCF, the default XGBoost model will only use peak width, prominence and temperature as inputs, while a model using width, prominence and altitude as inputs will be an alternative for cases when temperature data is unavailable."

and at line 531:

"The G22-Christchurch model and algorithm will be incorporated in the next version of ALCF, so that future work can apply this retrieval technique to other lidar and ceilometer datasets."

Minor comments:

Line 15: There's actually not only collision freezing. I suggest stating that INP needs to be involved in heterogeneous nucleation of ice at temperatures between 0°C and -40°C. Citation of Hoose and Möhler (2012, https://doi.org/10.5194/acp-12-9817-2012) should do the job.

R2C10: The revised manuscript now reads on line 14:

"Heterogeneous nucleation of ice in clouds occurs between -40 °C and 0 °C when SLW droplets interact with ice nucleating particles (INPs) such as dust and other aerosols, or other ice particles (Hoose and Möhler, 2012)."

Line 94: really just 21? I guess it's 61, isn't it?
https://www2.mmm.ucar.edu/rt/amps/information/configuration/configuration.html

R2C11: For the NZ grid used in this study, there are 21 levels.

Line 129: Existence of ice at T above 0°C. The statement is actually not quite true. Melting of ice depends on the dewpoint temperature. It can well exist at higher temperatures as long as dewpoint temperature (wet-bulb or ice-bulb temperature) is below 0°C. E.g., https://doi.org/10.1175/JAS-D-20-0353.1  Thus the WLCC liquid water statement should be treated somewhat relative.

R2C12: It is correct that ice can exist at temperatures above 0°C before the onset of melting. This section was rephrased and the line " … since ice cannot exist above 0°C" was removed. See also R1C12.

Frequently, the term 'cloud' appears at positions where it should actually be placed in plural form (e.g., lines 16, 29, 48). Do variants of the English language exist, where 'cloud' is both, plural and singular? Or are these just typos?

R2C13: These examples sounded natural to the author, but were nonetheless changed to avoid ambiguity.

Lines 183 and 250: use citep

R2C14: Fixed.